# Causal Discovery from Event Sequences by Local Cause-Effect Attribution

**Joscha Cüppers**°
CISPA Helmholtz Center
for Information Security
joscha.cueppers@cispa.de

**Sascha Xu**°
CISPA Helmholtz Center
for Information Security
sascha.xu@cispa.de

**Ahmed Musa**•
Institute for Structural Analysis
TU Dresden
ahmed.musa@tu-dresden.de

**Jilles Vreeken**
CISPA Helmholtz Center
for Information Security
jv@cispa.de

## Abstract

Sequences of events, such as crashes in the stock market or outages in a network, contain strong temporal dependencies, whose understanding is crucial to react to and influence future events. In this paper, we study the problem of discovering the underlying causal structure from event sequences. To this end, we introduce a new causal model, where individual events of the cause trigger events of the effect with dynamic delays. We show that in contrast to existing methods based on Granger causality, our model is identifiable for both instant and delayed effects.

We base our approach on the Algorithmic Markov Condition, by which we identify the true causal network as the one that minimizes the Kolmogorov complexity. As the Kolmogorov complexity is not computable, we instantiate our model using Minimum Description Length and show that the resulting score identifies the causal direction. To discover causal graphs, we introduce the CASCADE algorithm, which adds edges in topological order. Extensive evaluation shows that CASCADE outperforms existing methods in settings with instantaneous effects, noise, and multiple colliders, and discovers insightful causal graphs on real-world data.

## 1 Introduction

Suppose we are considering a multivariate event sequence. What caused a specific event to happen? Which variables are causes of each other? Data-driven methods can infer causal relationships from observed data. Existing methods for discovering causal networks from event sequence data [1–3] are based on *Granger causality* [4]. This purely predictive notion defines a variable $X$ to be a cause of another variable $Y$ if the past of $X$ helps to predict the future $Y$. It is a relatively weak notion of causality that excludes instantaneous effects and is often unable to discover true causal dependencies; in Granger causality, baking a cake is causal to a birthday.

In this paper, we instead build upon Pearl's model of causality, which assumes the existence of an an underlying causal structure in the form of a directed acyclic graph (DAG) [5]. In our context, such a graph describes the causal relationships between types of events, such as alarms in a network. We propose a new causal model for event sequences based on a one-to-one *matching* of individual events, where we model the process of one individual event of a certain type possibly causing an

---

°Equal contribution
•This work was done while at CISPA Helmholtz Center for Information Security

38th Conference on Neural Information Processing Systems (NeurIPS 2024).

individual event of another type. In our model, we take into account the uncertainty of whether an event is actually caused or independently generated, the uncertainty of an event actually causing an effect or failing to do so, and the uncertainty of the delay between cause and effect. As we will show, our model has several advantages, such as a clear notion of what event caused another and the identifiability for both instant and non-instant effects.

We base our theory on the Algorithmic Markov Condition (AMC) [6], which postulates that the true causal model achieves the lowest Kolmogorov complexity. As Kolmogorov complexity is not computable, we instantiate it via the Minimum Description Length (MDL) principle [7]. We show that our score is consistent, identifies the true causal direction for both instantaneous and delayed effects, and formally connect it to Hawkes processes. To discover causal networks in practice, we introduce the CASCADE algorithm, which adds edges in topological order. Through extensive empirical evaluation, we show that CASCADE performs well in practice and outperforms the state of the art by a wide margin. On synthetic data, CASCADE recovers the ground truth without reporting spurious edges, and on real-world data, it returns graphs that correspond to existing knowledge.

## 2 Preliminaries

We write $i \rightarrow j$ when $S_i$ is a cause of $S_j$ and $pa(j)$ for the set of parents of node $j$. We assume faithfulness, sufficiency, and the causal Markov condition [8].

**Information-Theoretic Causal Discovery**  The Algorithmic Markov Condition (AMC) postulates that the factorization of the joint distribution according to the true causal network achieves the lowest Kolmogorov complexity [6]. The Kolmogorov complexity $K(x)$ of a binary string $x$ is the length of the shortest program $p$ for a universal Turing machine $\mathcal{U}$ that computes $x$ and halts [9]. For a distribution $P$, it is the length of the shortest program that uniformly approximates $P$ arbitrarily well,

$$K(P) = \min_{p \in 0,1^*}\{|p| : \forall_y |\mathcal{U}(p, y, q) - P(y)| \leq \frac{1}{q}\} .$$

The AMC states that the Kolmogorov complexity of the joint distribution $P(X)$ is the sum of the complexities of the conditional distributions $P(X_i|pa(i))$ of the true DAG $G^*$, i.e.

$$K(P(X)) = \sum_{i=1}^{p} K\left(P(X_i|pa(i))\right) ,$$

up to a constant independent of the input. Due to, among others, the halting problem, Kolmogorov complexity is not computable, but we can approximate it from above. A statistically well-founded way to do so is by Minimum Description Length (MDL) [7, 10]. For a fixed class of models $\mathcal{H}$, MDL identifies a description length $L$ of encoding data $X$ together with its optimal model,

$$L(X \mid \mathcal{H}) = \min_{h \in \mathcal{H}}\left(L(X \mid h) + L(h)\right) .$$

Next, we introduce the assumed data generating process, its corresponding model class $\mathcal{H}$ and encoding length function $L$, and show under which conditions it can be identified.

## 3 Theory

To be able to infer causal relationships from observational data, we need to make assumptions about the underlying data-generating process [5]. The key assumption we make here is that an individual event of type $i$ at time $t$ with probability $\alpha_{i,j}$ causes an individual event of type $j$ at time $t' \geq t$. To illustrate, we give a toy example in Fig. 1 in which event sequence $S_i$ causes event sequence $S_j$. The individual events in $S_i$ occur uniformly at random. The first and third events in $S_i$ cause events in $S_j$, resp. with a delay of 0.2 and 0.3. The other two events in $S_i$ do not cause events in $S_j$, denoted by a delay of $\infty$. The final event in $S_j$ is due to noise, marked by $N_j$.

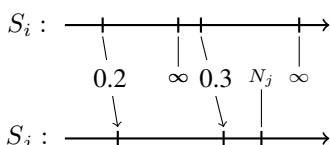

Figure 1: Cause-effect matching, where $S_i$ causes $S_j$.

Next, we formally describe the causal mechanism. We differentiate between source and effect nodes.

*Source* nodes are nodes $i$ in $G^*$ with an empty parent set $pa(i)$. For source nodes $i$ we assume that the events in $S_i$ occur uniformly at random with a rate of $\lambda_i$ events per time unit. This mechanism, commonly known as a homogeneous *Poisson process*, is used, for example, as a model for accident rates requiring hospital admission [11]. In this work, we focus on the delay times between individual events, denoted as $d_k$ for the delay $t_k - t_{k-1}$ and as $\Delta_{i \to i} = \{d_k\}_{k=1}^{n_i}$ for the sequence. For a Poisson process, the delay times are independently and exponentially distributed. Thus, we model a source event sequence $S_i$ as

$$S_i = \{t_k\}_{k=1}^{n_i}, \text{ where } t_k = \sum_{l=1}^{k} d_l, \quad \Delta_{i \to i} = \{d_k \sim \text{Exp}(\lambda_i) \, iid\}_{k=1}^{n_i}. \tag{1}$$

*Effect* nodes are nodes $j$ in $G^*$ with at least one parent $pa(j)$. For each effect node $j$, the individual events in $S_j$ are either caused by an individual event in an $S_i$ with $i \in pa(j)$ or due to noise. That is, reasoning from the causing node $i$, every event $t_k \in S_i$ may trigger an event of the effect $S_j$ with a probability of $\alpha_{i,j}$. If triggered, an individual event in $S_j$ will occur after a random delay $d_k$, drawn from a cause-effect specific delay distribution $\Phi_{i,j}$ parameterized by $\theta_{i,j}$, e.g. the rate $\lambda$ of an exponential distribution, and $\alpha_{i,j}$. If no event is triggered, then we model the delay as infinite, i.e. $d_k = \infty$. The sequence of delays $\Delta_{i \to j}$ from $S_i \to S_j$ is modeled as

$$\Delta_{i \to j} = \{d_k\}_{k=1}^{n_i}, \qquad d_k \sim \Phi_{i,j}(\alpha_{i,j}, \theta_{i,j}) \, iid, \qquad \phi_{i,j}(d) = \begin{cases} 1 - \alpha_{i,j} & \text{if } d = \infty \\ p(d; \theta_{i,j}) \cdot \alpha_{i,j} & \text{else} \end{cases} \tag{2}$$

where $\phi_{i,j}(d)$ denotes the density of the delay distribution. Thus, given the event sequence $S_i$ of the cause and delays $\Delta_{i \to j}$, the individual events in $S_j$ caused by $S_i$ are obtained by adding the delays $d_k$ to the time stamps $t_k$ of the individual cause events, with the reconstruction function $f$ as

$$f_{i,j}(S_i, \Delta_{i \to j}) = \{t_k + d_k \mid d_k \neq \infty\}, \text{ for } k = 1, \dots, n_i .$$

In addition, individual events in $S_j$ can also be due to noise. Like for source nodes, we assume these a Poisson process as per Eq. (1) with rate $\lambda_j$, i.e. $N_j \sim \text{Poisson}(\lambda_j)$. Putting this together, given causal structure $G^*$, an effect event sequence $S_j$ is generated by taking the union of the individual delays from the causal parents $pa(j)$ and the time stamps due to noise $N_j$,

$$S_j = \left( \bigcup_{i \in pa(j)} f_{i,j}(S_i, \Delta_{i \to j}) \right) \cup N_j . \tag{3}$$

Next, we instantiate an MDL score for this causal model and consider its identifiability.

### 3.1 Minimum Description Length Instantiation

We now develop a score for our causal model using MDL [7]. It consists of the cost of the data given the model, $L(S \mid \Theta)$, i.e. the negative log-likelihood of the data, and the cost of the model, i.e. that of the parameters, $L(\Theta)$, and that of the graph, $L(G)$, all measured in bits.

**Data Cost** The cost of data in bits directly corresponds to its negative log-likelihood, i.e. the likelihood of each delay as per Eq. (2) over all the event sequences corresponding to the parents of node $j$ and that of the noise events. Formally, we have

$$L(S_j \mid S_{pa(j)}, \Theta) = \sum_{i \in pa(j)} \sum_{d_k \in \Delta_{i \to j}} -\log(\phi_{i,j}(d_k)) + \sum_{d_l \in \Delta_{j \to j}} -\log(\phi_{j,j}(d_l)) .$$

The first term encodes those events that were caused by the parent $S_i$ through the delays $\Delta_{i \to j}$. Here, we use a Shannon-optimal coding that requires $-\log(\phi_{i,j}(d_k))$ bits per sample [7]. In the second term, we encode all remaining events as noise using the delay distribution of a Poisson process. For source events, i.e. variables without any parents, only the noise term is present.

The cost of all sequences is then simply $L(S \mid G, \Theta) = \sum_{j \in [p]} L(S_j \mid S_{pa(j)}, \Theta)$.

**Parameter Cost** Next, we define the costs of the DAG, $L(G)$, and that of the parameters, $L(\Theta)$. We encode the DAG in topological order. Per node we encode its number of parents $|pa(i)|$ and identify which those are, i.e. $L(G) = \sum_{k=0}^{d-1} \left( \log(k) + \log \binom{k}{|pa(i)|} \right)$. Depending on their type, we encode the parameters $\theta \in \Theta$. For parameters $\theta \in \mathbb{N}$ we use $L_{\mathbb{N}}$, the MDL-optimal encoding for integers [12]. It is defined as $L_{\mathbb{N}}(z) = \log^* z + \log c_0$ where $\log^* z$ is the expansion, $\log z + \log \log z + \cdots$ in which we only include positive terms. To ensure this is a valid encoding, i.e. one that satisfies the Kraft inequality, we set $c_0 = 2.865064$ [12]. For parameters $\theta \in \mathbb{R}$ we use $L_{\mathbb{R}}(\theta) = L_{\mathbb{N}}(d) + L_{\mathbb{N}}(\lceil \theta \cdot 10^d \rceil) + 1$ as the number of bits needed to encode a real number up to a user-specified precision [7]. For an edge $i \to j$, the parameters are the trigger probability $\alpha_{i,j}$ and those of the delay distribution $\phi$. For the cost of an edge we hence have $L(i \to j) = L_{\mathbb{R}}(\alpha_{i,j}) + \sum_{\theta \in \phi_{i,j}} L(\theta)$. For $\Theta$ as a whole, we have $L(\Theta) = \sum_{i \to j \in G} L(i \to j)$.

The overall MDL score is then

$$L(S \mid G, \Theta) + L(G) + L(\Theta) .$$

## 3.2 Identifiability

We now study the identifiability guarantees of our model and score, i.e. under what conditions we can identify from a given pair which is the cause and which the effect. Consider a pair of event sequences $S_i$ and $S_j$, where $S_i \to S_j$ and the cause $S_i$ is a source event while $S_j$ is an effect event.

**Instant Effects** We begin with the case of instant effects only. Instant effects are observed when the sampling frequency of the data, e.g. a daily time scale, is insufficient to pick up a difference in time, such as a financial crash that can spread across the globe within hours. It is well-known that Granger causality cannot identify the causal direction for instant effects [13]. In Pearl's causal framework, on the other hand, the causal direction between two binary variables is identifiable [14–16]. We can build upon these results and show that our causal model and MDL-based score can identify the causal direction for non-deterministic instant effects.

**Theorem 1.** *Let $S_i$ be an event sequence generated by a Poisson process as per Eq. (1) and $S_j$ be an effect of $S_i$ as per Eq. (3), with, low noise $\lambda_j < (1 - \alpha_{i,j})\lambda_i$, and a trigger probability $\alpha_{i,j} < 1$.*

*In the case of exclusively instant effects, i.e. $\phi_{i,j}(d) = \delta(d)$, where $\delta(d)$ is the Dirac delta function, the MDL score in the true causal direction is lower than in the anti-causal direction, i.e.*

$$\lim_{n_i \to \infty} L(S_j \mid S_i, \Theta_1) + L(S_i \mid \Theta_1) < L(S_i \mid S_j, \Theta_2) + L(S_j \mid \Theta_2) .$$

We provide the full proof in the Appx. A.1, the general idea is under a non-deterministic trigger mechanism, i.e. $\alpha_{i,j} < 1$. Then, in the causal direction, we can fully explain $S_j$ with $S_i$, but not vice-versa, as the cause is generated by a Poisson process. If $\alpha_{i,j} = 1$, i.e. the process is deterministic, we always observe cause and effect together, making them indistinguishable.

**Delayed Effects** Next, we consider the case of exclusively delayed effects. Here, there is an inherent asymmetry in the benefit of knowing the cause versus the effect. As shown by Didelez [17] for marked point processes, and later used by Xu et al. [1], Eichler et al. [18] for Granger causality in Hawkes processes, the intensity of observing the cause after an event of the effect is unchanged. That is, the future of the cause is independent of the past of the effect, while if a cause triggers an effect, the intensity of the effect is increased by the cause. We have the following identifiability guarantee.

**Theorem 2.** *Let $S_i$ be an event sequence generated by a Poisson process as per Eq. (1) and $S_j$ be an effect of $S_i$ as per Eq. (3), such that $H(\phi_{j,j}) > H(p(; \theta_{i,j})) + \alpha_{i,j}^{-1} H(\mathcal{B}(\alpha_{i,j})) + \alpha_{j,j}^{-1} H(\mathcal{B}(\alpha_{j,j}))$, where $H$ denotes the entropy and $\mathcal{B}$ the Bernoulli distribution.*

*Then the matching in the anti-causal direction $\Delta_{j \to i}$ of the effect $S_j$ to the cause $S_i$ has a worse MDL score than the true matching $\Delta_{i \to j}$, i.e.*

$$L(S_j \mid S_i, \Theta_{i \to j}) + L(S_i \mid \Theta_i) < L(S_i \mid S_j, \Theta_{j \to i}) + L(S_j \mid \Theta_j) .$$

We provide the full proof in the Appx. A.2. In the anti-causal direction $S_j \to S_i$, the delay times follow the same exponential distribution of $\text{Exp}(\lambda_i)$, leading to no gain in score compared to the

self-delay encoding. On the other hand, in the true causal direction, knowing the times of the cause leads to a better knowledge of the delay and hence a lower cost, so long as the delay distribution $\phi_{i,j}$ provides a better description than treating it as noise. This requirement is closely related to the algorithmic Markov condition, which postulates that the shortest description of a variable is given through its parents.

### 3.3 Connection to Hawkes Processes

Hawkes processes [19] are analytically convenient and well-suited for modeling real-world processes where events trigger further events, e.g. earthquakes triggering aftershocks. Consequently, the majority of methods focusing on Granger causality are based on Hawkes processes [1–3]. The Hawkes process extends the Poisson process by incorporating the influence of past events on the intensity, i.e. the rate of occurrence of future events. This is done by means of excitation functions $v_{i,j}(t - t_k)$, which increase/inhibit the intensity of future events based on past events. The intensity function of a Hawkes process under a DAG structure is given by

$$\lambda_j(t) = u_j + \sum_{i \in pa(j)} \sum_{t_k < t, t_k \in S_i} v_{i,j}(t - t_k) \ .$$

Each event $t_k \in S_i$ increases the intensity of seeing an effect by $v_{i,j}(t - t_k)$. The main difference between our model and a Hawkes process is our direct trigger model from cause to effect. In a Hawkes process, an event of type $i$ increases the intensity and, therewith, the probability of effect events occurring. That is, contrary to our framework, in a Hawkes process there is no explicit one-to-one relationship between causing and effect events, i.e. no one event can be attributed solely to causing another. Nonetheless, in Appendix A.5 we show how to identify $S_i$ as a parent of $S_j$ by constructing a sequence of delays $\Delta_{i \to j}$ with the most-influential past event and therewith $\phi_{i,j}$. If $\phi_{i,j}$ fulfills Theorem 2, we can identify $S_i$ as a parent of $S_j$. Hence, should the data be generated by a Hawkes process, our method can still pick up the causal relationship between the two event classes, so long as there are sufficiently many events where $S_i$ is the primary cause.

## 4 Algorithm

With our model in place, we now turn to the problem of discovering the underlying causal structure from an observed sequence of events. In recent years, several methods that find and proceed on a topological ordering of the true graph have been introduced [20–22], which outperform other score-based frameworks such as GES [23] in terms of accuracy. We here propose the CASCADE algorithm that instantiates this idea for information-theoretic scores. We prove that in the limit, it recovers not only the correct topological ordering but also the correct parent set of each node.

CASCADE derives its guarantees from the *gain* in bits of adding an edge $i \to j$ to the model, i.e.

$$g(i \to j \mid \Theta) = L(S_j \mid S_{pa(j)}, \Theta) - L(S_j \mid S_{pa(j) \cup i}, \Theta \cup \theta_{i,j}) + L(i \to j) \ .$$

The edge cost $L(i \to j)$ is constant and independent of the number of samples $n_i$. In the limit $n_i \to \infty$, the gain inherits the identifiability guarantees from Sec. 3.2, such that $g(i \to j \mid \Theta) > g(j \to i \mid \Theta)$ if $S_i$ is a true ancestor of $S_j$. In other words, the gain of an edge is greater in the causal than in the anti-causal direction.

### 4.1 High Level Overview

CASCADE initializes the model $\Theta$ with an empty graph $G$ and without any causal edges. During the search, we maintain a set of nodes $C = [p]$, from which we remove nodes in a topological order of $G^*$. We iterate over the following four steps until $C$ is empty.

1. **Source Node Selection**: Select that node $i \in C$ with minimal gain for any edge $j \to i$, $j \in C$, i.e.

$$\underset{i \in C}{\arg \min} \ \underset{j \in C}{\max} \ g(j \to i \mid \Theta) - g(i \to j \mid \Theta) \ . \tag{4}$$

2. **Edge Adding**: Add all *outgoing* edges from $i \to j$, $j \in C$, to $G$ that *improve* our score.

3. **Edge Pruning**: Remove all *incoming* edges $j \to i$ from $G$ that harm our score.

4. **Node Set Update** Remove $i$ from $C$.

Each iteration, CASCADE selects that node $i$, which has the minimal achievable gain when adding any edge $j \rightarrow i$ to the current graph $G$, expressed in Eq. (4); below, we will show that under our causal model this node is guaranteed to be a true source of the graph $G^*$. We then add all edges from $i$ to nodes $j \in C$ that improve our score; provided that all true causal edges $i \rightarrow j$ were added, there is now at least one node $j \in C$ whose parents are all accounted for, that in the next iteration can be identified as a source. We remove edges $j \rightarrow i$ from $G$ to remove shortcuts. By repeating this process, CASCADE proceeds in a topological order of the true graph $G^*$. In total, CASCADE requires $p$ iterations, leading to an overall cubic complexity $O(p^3)$.

**Source Node Selection.** To identify a source node in the graph, we can use the identifiability guarantees from Sec. 3.2. They show that the gain $g(i \rightarrow j \mid \Theta)$ correctly orients the edge $i \rightarrow j$ in the unconfounded bi-variate case. We additionally require that the edge gain is *pathwise oracle*, i.e. it can identify the direction of the path from $i$ to $k$.

**Theorem 3.** *Given an event sequence $S$ generated by a causal structure $G^*$, let $S_i$ be a source node of $G^*$ and $S_v$ be a descendant of $S_i$, where there exists a path $i \rightarrow j \rightarrow \cdots \rightarrow v$ in $G^*$.*

*Then, the gain in the causal direction of the path $g(i \rightarrow v \mid \Theta) - g(v \rightarrow i \mid \Theta)$ is greater.*

We provide the proof in Appx. A.3. We can now show that the criterion in Eq. (4) selects nodes in a topological ordering of $G^*$. Initially, CASCADE has to identify a true source of $G^*$, i.e. a node $i$ without parents. For that node $i$, all other nodes $j$ are either ancestors or independent of $i$. If $i$ is an ancestor of $j$, then $g(j \rightarrow i \mid \Theta) - g(i \rightarrow j \mid \Theta) < 0$, i.e. the gain in the anti-causal direction is lower. If $i$ is independent of $j$, then $g(j \rightarrow i \mid \Theta) = 0$ and $g(i \rightarrow j \mid \Theta) = 0$. Hence, the maximum achievable gain for a node without parents is zero.

Now consider a node $v$ which does have a parent. For this node, there exists an ancestor $u$ which is a true source. Hence, for that pair $g(u \rightarrow v \mid \Theta) - g(v \rightarrow u \mid \Theta) > 0$. Consequently, the maximum achievable gain is positive, whilst for a source node, we can maximally achieve zero, allowing us to identify true sources with Eq. (4).

In the next step, we add all outgoing edges from the source $i$ to $G$ that improve the score. As $G^*$ is a DAG, we are now guaranteed to have another node $j$, whose incoming edges are all accounted for in $G$. Then, as per the causal model from Eq. (3), the only events that remain are those of the noise $N_j$. Hence, $j$ is now a source node for which the guarantees from above apply. By repeating this process, CASCADE thus follows a topological order of $G^*$.

**Edge Addition** Given a source $i$, CASCADE adds all outgoing edges $i \rightarrow j$ that improve the score. We restrict the set to nodes $j \in C$ from the candidate set only, i.e. to nodes further down the topological order. By the Algorithm Markov Condition, the description length of the true set of parents of a node $j$ is smaller than the description length of any other set of parents, and hence the gain of the true edge is positive in the limit of $n_i \rightarrow \infty$.

When adding an edge $i \rightarrow j$, where there is already an edge $v \rightarrow j$, we use an Expectation Maximization approach to attribute all events to their respective cause. That is, we first find the bi-variate alignment $\Delta_{i \rightarrow j}$ using all events in $S_j$. Now, it is very likely that there are conflicts between $\Delta_{i \rightarrow j}$ and $\Delta_{v \rightarrow j}$, as the same event can be attributed to both $i$ and $v$. In those cases, we choose that event where the density $\phi_{i,j}/\phi_{v,j}$ is higher and set the delay to infinity in the other matching. After re-assigning all events, we refit the delay distribution function $\phi_{i,j}$ using the new matching.

**Edge Pruning** Lastly, we deal with removing any shortcuts that have been added in the previous iteration. With the previous two steps, we are guaranteed to have a superset of all true causal edges incoming to $i$. Fortunately, we can prune such edges directly with MDL by removing any incoming edge $i \rightarrow j$ that does not improve the MDL score. In the chain graph $i \rightarrow j \rightarrow v$, we would remove $i \rightarrow v$ as the edge $j \rightarrow v$ is sufficient to explain the data. In practice, given the current set of parents of $i$ in $G$, we search for the true set of parents by starting with the empty set and greedily adding only those edges that improve the score.

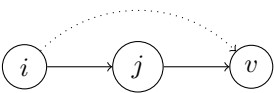

Figure 2: Causal chain

As we show in Appx. A.3, a shortcut always has a lower gain than the true edge and hence will not be

re-added. In this manner, we are asymptotically left with only the true causal parents. We can now finally show the consistency of CASCADE.

**Theorem 4.** *Given an event sequence $S$, where each individual subsequence $S_i$ was generated as per Eq. (3) by an underlying causal graph $G^*$. Assuming all $\Delta_{i \to j}$ are the true causal matchings. Under the Algorithmic Markov Condition, CASCADE recovers the true graph $G^*$ for $n \to \infty$.*

We postpone the proof to Appx. A.7. In the experiment section, we show that CASCADE recovers the true DAG even in challenging settings and works well on real-world data.

## 5 Related Work

Causal discovery on observational data is an active research topic. Two main research directions exist: constraint-based [5] and score-based [23, 24] methods. Our approach belongs to the latter and is based on the Algorithmic Markov Condition [6]. While Kolmogorov complexity is uncomputable, Marx and Vreeken [10] formally showed that if we instantiate the AMC with two-part MDL [7], we, on expectation, achieve the same results. MDL has been successfully used for bivariate causal inference [25, 15, 26], causal discovery [27], identifying hidden confounding [28], identifying mechanisms shifts [29], and identifying selection bias [30].

In this paper, we consider point processes. Particularly close to our method are Hawkes processes [31] as a way to model the influence of past events onto future events. As such, our work is also related to the concept of transfer entropy [32], which measures the influence in terms of Shannon entropy. Budhathoki and Vreeken [33] proposed an MDL-based method for bivariate causal inference on event sequences, which is unsuitable for learning a global causal structure.

Existing methods for discovering causal graphs from event sequence data focus on different instantiations of Granger causality and can mostly be categorized by different intensity functions. Most common are parametric approaches with different regularizing [34, 2, 1]. ADM4 [34] uses the nuclear matrix norm in combination with lasso, THP [2] uses BIC for regularization. The method MDLH by Jalaldoust et al. [3] is most closely related, as they also use MDL for regularization. NPHC [35] takes a non-parametric approach by using a moment matching method to fit second and third-order integrated cumulants. A recent development is neural point processes. Mei and Eisner [36] propose a deep neural network that learns the dependencies [36], which Xiao et al. [37] extended to include attention mechanisms. Zhang et al. [38] first learn a neural point process and then use a feature importance attribution method to obtain a weight matrix of pairwise variable influence.

## 6 Experiments

We evaluate CASCADE on both synthetic and real-world data. CASCADE is implemented in Python. We provide the source code, along with the synthetic data generator and the used real-world datasets online.[3] We compare our method to four of state of the art methods: THP [2] as representative for the regularized parametric approaches, CAUSE [38] as representative for the neural point processes and NPHC [35] as a representative non-parametric approach, and MDLH [39] who also rely on MDL, as our most closely related competitor. CAUSE and NPHC do not return a graph but rather a weight matrix where the weight indicates the strength of the causal relation. On synthetic data, we can obtain a graph by thresholding such that we optimize the $F1$ score.

### 6.1 Evaluation

We evaluate the estimated graphs in terms of structural similarity by the Structural Hamming Distance (SHD) [40], in terms of causal similarity by the Structural Intervention Distance (SID) [41], and predictive performance by F1 score. To compare graphs of different sizes, we report the scores normalized by the maximally achievable SHD/SID and show the unnormalized scores in Appendix B.

NPHC, CAUSE, and MDLH can and often do return cyclic graphs. As SID is strictly only defined for acyclic graphs, we omit these methods from the SID evaluation.

---

[3]`https://eda.rg.cispa.io/prj/cascade/`

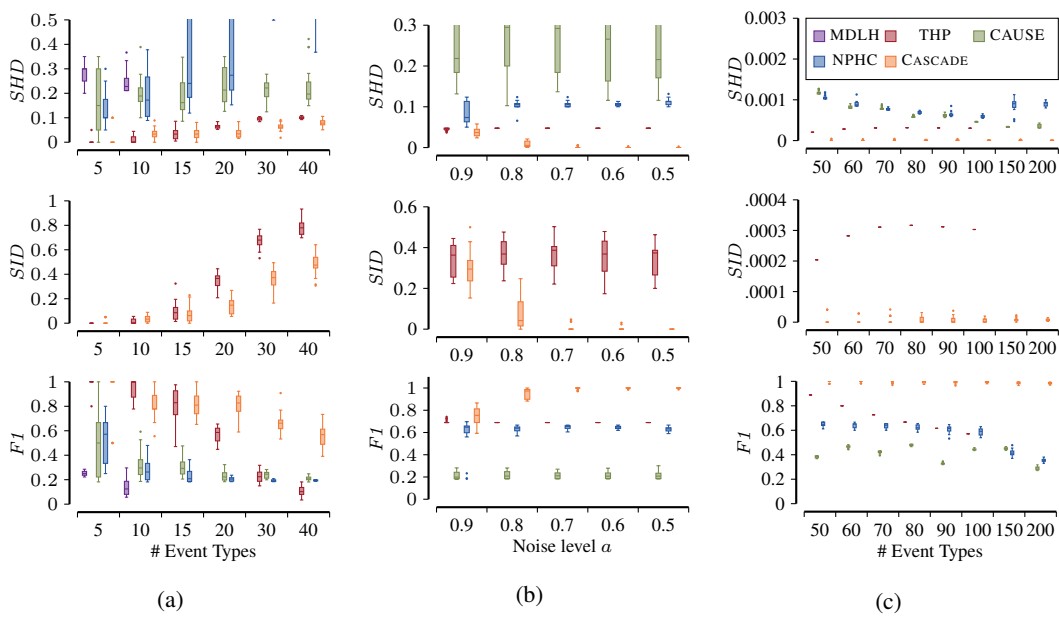

Figure 3: DAG recovery in different settings. We show normalized SHD, normalized SID, and $F1$ score, the $Y$-axis are truncated for better visualization. In (a) we vary the number of event types, on the SID score we observe that the graph reported by CASCADE is casually, the most similar to the true DAG. In (b) we decrease the noise, CASCADE does recover a close causal graph, even under high noise. Finally, in (c) we increase the number of parents of a collider, we observe that a high number of parents does not pose a problem for CASCADE.

## 6.2 Synthetic Data

We begin by comparing all methods on data with known ground truth. To this end, we generate synthetic data. We generate both data within and outside our causal model and vary aspects such as noise intensity, number of event types and the number of parents of a variable. We describe the full data-generating process in Appendix B.

**Sanity Check** We start with a sanity check on data without any structure over 20 variables, CASCADE correctly does not report any causal edge. THP reports in 45% of the cases at least one spurious edge. We omit the results of CAUSE and NPHC as it is unclear how to choose a meaningful, non-trivial threshold, in this setting. MDLH did not terminate within 96 hours.

**Scalability** We evaluate how well each method scales under an increasing number of variables. We vary the number of nodes, which correspond to the number of unique event types, from 5 to 50 and report the results in Fig. 3a. As MDLH did not terminate within 96 hours for 15 variables, we omit it from here on out. For a lower number of nodes, both CASCADE and THP obtain far better results than NPHC and CAUSE. With increasing event types, all methods SID and F1 scores decrease. Amongst all methods, CASCADE scales best with an increasing number of nodes, whereas Granger causality based methods such as THP and NPHC find many spurious edges of connected but not causal variables. On the other hand, CASCADE is the most accurate method for a higher number of nodes, showing the efficacy of its causal model and MDL-based approach.

**Noise** Next, we assess the impact of noise, which are events that is not caused by any parent. To this end, we vary the 'cause' probability and the fraction of events due to additive noise. We do so by varying a noise parameter $a$, adding an additional $n_i \cdot a$ events to $S_i$ (additive noise), and by setting the 'cause' probability $\alpha = 1 - a$, i.e. we decrease additive noise and increase trigger probability. We show the results in Fig. 3b. We observe that CASCADE does quite well for high noise and that for noise levels of $a = 0.7$ and lower, it (mostly) recovers the true DAG. All other methods perform considerably worse.

**Colliders**  Matching an effect event to the correct parent, resp. modeling the correct excitation, becomes increasingly challenging for a larger number of parents. We test this through a setting where half ($\lceil \frac{p-1}{2} \rceil$) the variables converge into a collider, and the other half ($\lfloor \frac{p-1}{2} \rfloor$) are independent. We vary the total number of variables, $p$ and we show the results in Fig. 3c. We observe that CASCADE achieves almost perfect results. THP is robust, but with an increasing number of nodes, it starts to miss edges. Beyond 100 variables, it does not terminate within 24 hours. To validate that our method can recover structures with multiple colliders, we repeat the same experiment where 10% of nodes are colliders. That is, for 50 event types, 5 are colliders and 23 direct causes of all 5 colliders. The remaining 22 are independent. Resulting in an $F1$ score of 0.97 for 50 event types, slightly decreasing to 0.82 for 200 event types; as such CASCADE can deal well with multiple colliders.

**Instantaneous Effects**  Next, we evaluate performance under instantaneous effects. First, we consider data with exclusively instant effects. CASCADE achieves an average unnormalized SHD of 32.8. The second best-performing method, NPHC, achieves 46.85. Next, we generate a setting where 90% of the effects are instantaneous and the others occur with a small delay. CASCADE improves to an SHD of 19.45, while NPHC achieves the second lowest average with 47.5. We provide all results in the Appendix B.

**Hawkes Processes**  Finally, we evaluate how effectively CASCADE recovers the true DAG on data generated by a Hawkes process. We vary the intensity of the excitation function, i.e., the expected number of events generated per cause. We show the results in Figure 4. We observe that CASCADE performs best when our assumptions hold, when there is one effect per cause or fewer, but still demonstrates strong performance across all settings.

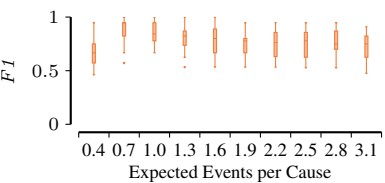

Figure 4: DAG recovery on data generated by a Hawkes process.

## 6.3   Real-World Data

We evaluate CASCADE on three distinct datasets of real-world event sequences. We begin by evaluating CASCADE on a dataset of network alarms, where the causal structure is known.

**Network Alarms**  This data was provided by Huawei for the NeurIPS 2023 CSL-competition[4] and consists of data from a simulated network of devices in which alarms can cause other alarms. We run all methods and get an (unnormalized) SHD score of 42 for CASCADE, 127 for THP, 214 for NPHC, and 1564 for CAUSE. As the network connectivity structure is known, we can take it into account during the search. THP supports this natively, CASCADE can be trivially constrained to only consider the given edges. CASCADE correctly identifies 142 out of 147 causal edges, THP 20. Neither method reports spurious edges. We show the full recovered graph in Appendix B.5.

**Global Banks**  Second, we run CASCADE on a daily return volatility dataset [42], we follow the preprocessing of Jalaldoust et al. [39], specifically we turn the time series into an event sequence by rolling a one year window over the data and register an event if the last value is among the top 10%. The dataset includes the 96 world's largest publicly traded banks. We show the largest discovered subgraph in Fig. 5. In addition, three unconnected sub-graphs are discovered, one covering two banks in Australia and two others connecting banks in Japan, which we provide in Appendix B.5.

**Daily Activities**  We run CASCADE on a dataset of recorded daily activities [43]. Our method reports plausible causal connections such as *Sleeping End → Showering Start → Showering End → Breakfast Start → Breakfast End*, etc. We show the complete graph in the Appendix B.5. This result reinforces the suitability of our causal model and CASCADE for real-world data, and illustrates the potential of our method to discover causal structures in a wide range of applications.

## 7   Conclusion

We studied the problem of causal discovery from event sequences, we propose a cause-effect matching approach to learn a fully directed acyclic graph (DAG). To this end, we introduced a new causal model

---

[4]https://github.com/huawei-noah/trustworthyAI/tree/master/competition/NeurIPS2023/sample

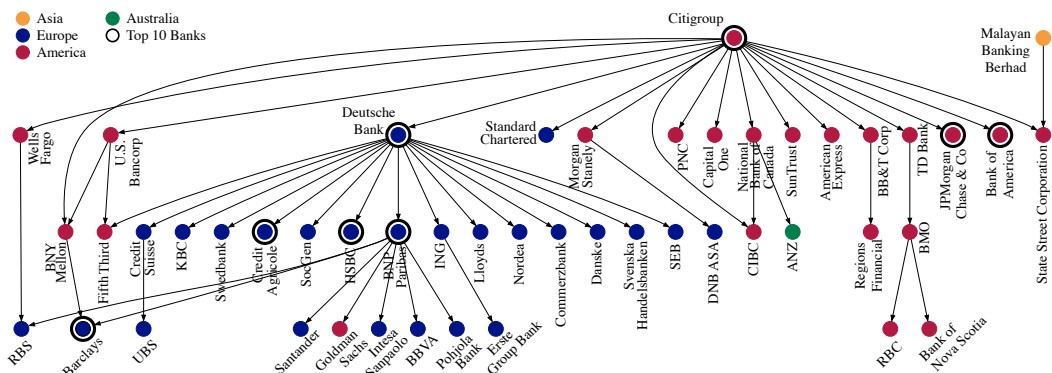

Figure 5: Result of CASCADE on the *Global Banks* dataset, we show the largest subgraph, we highlight the 10 largest, by assets, banks. We clearly see CASCADE recovers locality and that larger banks have a strong influence on the market, both information not provided in the input.

and an MDL based score. We proposed the CASCADE algorithm to discover causal graphs through a topological search from observational data. Finally, we evaluated CASCADE on synthetic and realistic data. On synthetic data, we find that CASCADE is either the best or close to the best-performing method across all settings, both within and outside our causal model. In particular, whenever conditions get challenging, e.g. due to noise or with multiple colliders, CASCADE outperforms all other methods by a significant margin. We examined how CASCADE performs on real world event sequences, where the true data-generating process may lie outside our causal model. We found that CASCADE recovers meaningful graphs that match with a common understanding of the world.

**Limitations** As is necessary, we have to make causal assumptions. The most prominent in our work is the direct matching between a cause event and an effect event – which precludes modeling of a single event causing multiple other events, as well as multiple events jointly causing a single effect event – and that we only consider excitatory effects – which precludes modeling the absence of events due to a cause. Our proof of identifiability for instantaneous effects depends on the strengths of the trigger resp. noise probabilities. The identifiability of the model seems provable via the independence of these, but how to operationalize this into an effective score and search algorithm are open questions.

**Future Work** Currently, our structural equations are 'or' relations over the parent's variables. An interesting future direction would be to explore 'and' relations, e.g., A and B together cause C. This raises several questions, like how close to each other A and B have to occur or if the order matters. Another interesting future direction is to allow matching of multiple causing events to one event, where each parent *could* have caused the event. This would allow us to answer counterfactual questions, such as if a causing event had not occurred, would we nevertheless observe its effect? This strongly relates to the firing squad example by Pearl [5], where multiple guards shoot a prisoner at the same time; if one guard did not shoot, the prisoner would still have died.

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

# A Theory

## A.1 Proof - Identifiability on Instant Effects

**Theorem 1.** *Let $S_i$ be an event sequence generated by a Poisson process as per Eq. (1) and $S_j$ be an effect of $S_i$ as per Eq. (3), with, low noise $\lambda_j < (1 - \alpha_{i,j})\lambda_i$, and a trigger probability $\alpha_{i,j} < 1$.*

*In the case of exclusively instant effects, i.e. $\phi_{i,j}(d) = \delta(d)$, where $\delta(d)$ is the Dirac delta function, the MDL score in the true causal direction is lower than in the anti-causal direction, i.e.*

$$\lim_{n_i \to \infty} L(S_j \mid S_i, \Theta_1) + L(S_i \mid \Theta_1) < L(S_i \mid S_j, \Theta_2) + L(S_j \mid \Theta_2) .$$

*Proof.* Let $n_i$ be the number of events in $S_i$ and $n_j$ the number of events in $S_j$. As $n_i \to \infty$

$$L(S_i) = n_i H(\phi_{i,i}), \qquad L(S_j|S_i) = n_i H(\mathcal{B}(\alpha_{i,j}))$$

where $\mathcal{B}$ is the Bernoulli distribution. In the reverse direction, we have

$$L(S_j) = n_j H(\phi_{j,j}), \qquad L(S_i|S_j) = n_i H(\mathcal{B}(\alpha_{i,i})) + (n_i - n_j)H(\phi_{i,i})$$

To show

$$L(S_i) + L(S_j|S_i) < L(S_j) + L(S_i|S_j)$$
$$n_i H(\phi_{i,i}) + n_i H(\mathcal{B}(\alpha_{i,j})) < n_j H(\phi_{j,j}) + n_i H(\mathcal{B}(\alpha_{i,i})) + (n_i - n_j)H(\phi_{i,i})$$

$\alpha_{i,j} = \alpha_{i,i}$ since every event that does not cause an i, can not be explain by j in the reverse direction, hence

$$n_i H(\phi_{i,i}) + \cancel{n_i H(\mathcal{B}(\alpha_{i,j}))} < n_j H(\phi_{j,j}) + \cancel{n_i H(\mathcal{B}(\alpha_{i,i}))} + (n_i - n_j)H(\phi_{i,i})$$
$$n_i H(\phi_{i,i}) < n_j H(\phi_{j,j}) + n_i H(\phi_{i,i}) - n_j H(\phi_{i,i})$$
$$\cancel{n_i H(\phi_{i,i})} < n_j H(\phi_{j,j}) + \cancel{n_i H(\phi_{i,i})} - n_j H(\phi_{i,i})$$
$$n_j H(\phi_{i,i}) < n_j H(\phi_{j,j})$$
$$H(\phi_{i,i}) < H(\phi_{j,j})$$

For $H(\phi_{i,i}) < H(\phi_{j,j})$ to hold $n_i > n_j$.

$$n_i \propto \lambda_i , \qquad n_j \propto \alpha_{i,j}\lambda_i + \lambda_j$$

$$\lambda_i > \alpha_{i,j}\lambda_i + \lambda_j$$
$$\lambda_i - \alpha_{i,j}\lambda_i > \lambda_j$$
$$(1 - \alpha_{i,j})\lambda_i > \lambda_j$$

It directly follows that $H(\phi_{i,i}) < H(\phi_{j,j})$, and hence for $n_i \to \infty$

$$L(S_j|S_i, \Theta_1) + L(S_i|\Theta_1) < L(S_i|S_j, \Theta_2) + L(S_j|\Theta_2) .$$

$\square$

## A.2 Proof - Identifiability on Delayed Effects

**Theorem 2.** *Let $S_i$ be an event sequence generated by a Poisson process as per Eq. (1) and $S_j$ be an effect of $S_i$ as per Eq. (3), such that $H(\phi_{j,j}) > H(p(; \theta_{i,j})) + \alpha_{i,j}^{-1} H(\mathcal{B}(\alpha_{i,j})) + \alpha_{j,j}^{-1} H(\mathcal{B}(\alpha_{j,j}))$, where $H$ denotes the entropy and $\mathcal{B}$ the Bernoulli distribution.*

*Then the matching in the anti-causal direction $\Delta_{j \to i}$ of the effect $S_j$ to the cause $S_i$ has a worse MDL score than the true matching $\Delta_{i \to j}$, i.e.*

$$L(S_j \mid S_i, \Theta_{i \to j}) + L(S_i \mid \Theta_i) < L(S_i \mid S_j, \Theta_{j \to i}) + L(S_j \mid \Theta_j) .$$

We will show that the delays between $S_i$ itself, i.e. $\Delta_{i \to i}$, and the delays between $S_j$ to $S_i$, i.e. $\Delta_{i \to j}$, are equivalent.

*Proof.* We consider a source event $S_i$ with exponentially distributed delays, i.e. $\Delta_{i \to i} \sim \exp(\lambda_i)$. Consider any event $t_k \in S_j$, then the intensity of observing an event in $S_i$ at time $t > t_k$ is given by $\lambda_i$. The distribution of the delay to the next event in $S_i$ is exponential with $\lambda = \lambda_i$. Thus, the difference between

$$L(S_i|S_j) - L(S_i) = \sum_{d_k \in \Delta_{i \to j}} \log(\lambda_i) - d_k/\lambda_i - \sum_{d_l \in \Delta_{i \to i}} \log(\lambda_i) + d_l/\lambda_i = 0 \ .$$

It remains to show that the likelihood in the causal direction is better when conditioning effect on the cause, i.e. $L(S_j|S_i) < L(S_j)$.

**Gain by $i \to j$** For $i$ to cause $j$ it has to provide information about $j$, that is the cost of selecting which $i$ events cause $j$, and with what dealys. Additional it has to ofset the cost which $j$ events do not have to be encoded as a self delay. Formally this is,

$$n_i H(\mathcal{B}(\alpha_{i,j})) + n_{i,j} H(p(; \theta_{i,j})) + n_j H(\mathcal{B}(\frac{n_{i,j}}{n_j})) < n_{i,j} H(\phi_{j,j}) \quad ,$$

where $n_{i,j}$ are the number of events in $j$ caused by $i$, and $H(p(; \theta_{i,j}))$ is the entropy of distribution described by the pdf $p$.

As $n \to \infty$,

$$L(S_j) = n_j H(\phi_{j,j}) \qquad L(S_j|S_i) = n_i H(\phi_{i,j}) + (n_j - n_{i,j}) H(\phi_{j,j}) + n_j H(\mathcal{B}(\alpha_{j,j}))$$

To show,

$$L(S_j) > L(S_j|S_i)$$

$$n_j H(\phi_{j,j}) > n_i H(\phi_{i,j}) + (n_j - n_{i,j}) H(\phi_{j,j}) + n_j H(\mathcal{B}(\alpha_{j,j}))$$

$$n_{i,j} H(\phi_{j,j}) + \underline{(n_j - n_{i,j}) H(\phi_{j,j})} > n_i H(\phi_{i,j}) + \underline{(n_j - n_{i,j}) H(\phi_{j,j})} + n_j H(\mathcal{B}(\alpha_{j,j}))$$

$$n_{i,j} H(\phi_{j,j}) > n_i H(\phi_{i,j}) + n_j H(\mathcal{B}(\alpha_{j,j}))$$

we can substitute $n_i H(\phi_{i,j}) = n_{i,j} H(p(; \theta_{i,j})) + n_i H(\mathcal{B}(\alpha_{i,j}))$

$$n_{i,j} H(\phi_{j,j}) > n_{i,j} H(p(; \theta_{i,j})) + n_i H(\mathcal{B}(\alpha_{i,j})) + n_j H(\mathcal{B}(\alpha_{j,j}))$$

Now, note that the number of caused items $\alpha_{i,j} n_i = n_{i,j}$ and $\alpha_{j,j} n_j = n_{i,j}$ , then it follows

$$n_{i,j} H(\phi_{j,j}) > n_{i,j} H(p(; \theta_{i,j})) + \frac{n_{i,j}}{\alpha_{i,j}} H(\mathcal{B}(\alpha_{i,j})) + \frac{n_{i,j}}{\alpha_{j,j}} H(\mathcal{B}(\alpha_{j,j}))$$

$$H(\phi_{j,j}) > H(p(; \theta_{i,j})) + \frac{1}{\alpha_{i,j}} H(\mathcal{B}(\alpha_{i,j})) + \frac{1}{\alpha_{j,j}} H(\mathcal{B}(\alpha_{j,j}))$$

Hence, we show that $i \to j$ is identifiable. $\qquad\qquad\square$

### A.3 Proof - Path Identifiability

**Theorem 3.** *Given an event sequence $S$ generated by a causal structure $G^*$, let $S_i$ be a source node of $G^*$ and $S_v$ be a descendant of $S_i$, where there exists a path $i \to j \to \cdots \to v$ in $G^*$.*

*Then, the gain in the causal direction of the path $g(i \to v \mid \Theta) - g(v \to i \mid \Theta)$ is greater.*

*Proof.* We begin by proving that the path identifiability holds for a triplet of nodes $i \to j \to v$, by constructing a new alignment from $i \to v$.

From $\Delta_{i \to j}$ and $\Delta_{j \to v}$ we can construct $\Delta_{i \to v}$. For each $d_k \in \Delta_{i \to j}$, there is a corresponding $d_l \in \Delta_{j \to v}$, i.e. the trigger time of the triggered event. To construct $\Delta_{i \to v}$, we consider the following cases:

1. If $d_k = \infty$ for $d_k \in \Delta_{i \to j}$, then $d_k \in \Delta_{i \to v}$, is set to $d_k = \infty$.

2. Let $d_l \in \Delta_{j \to v}$ be the delay of event $a$ of type $j$ where $a$ has been caused by delay $d_k$. If $d_k \neq \infty$ for $d_k \in \Delta_{i \to j}$ and $d_l = \infty$ then then $d_k \in \Delta_{i \to v}$, is set to $d_k = \infty$.

3. Let $d_l \in \Delta_{j \to v}$ be the delay of event $a$ of type $j$ where $a$ has been caused by delay $d_k$. If $d_k \neq \infty$ for $d_k \in \Delta_{i \to j}$ and $d_l \neq \infty$ then then $d_k \in \Delta_{i \to v}$, is set to $d_k = d_k + d_l$.

As $\Delta_{i \to v}$ is another valid alignment, and $i$ remains a source node, the guarantees of Theorem 1 and 2 hold, i.e. the path is identifiable. This extends to a path of arbitrary length. Consider an additional edge $v \to w$, then we construct the alignment $\Delta_{i \to w}$ by considering the delays of $\Delta_{i \to v}$ and $\Delta_{v \to w}$.

Furthermore, we can show that the true path $j \to v$ has a better gain than the shortcut $i \to v$, so that we can remove the shortcut in the pruning stage. (1) Since $i \to j$ and $j \to v$ are independent processes, it follows either $\mathrm{Var}(\phi_{i,v}) > \mathrm{Var}(\phi_{j,v})$, or $\alpha_{i,v} > \alpha_{j,v}$ and by that a more costly description of $v$.

(2) If $j \to v \notin G$ , we can construct a new function $f_{i,v} = f_{j,v}(f_{i,j}(S_i, \Delta_{i \to j}) \cup N_j, \Delta_{j \to v})$ by Theorem 2 it follows that edge $i \to v$ improves our score.

(3) Assume $j \to v \in G$ then each individual $v$ event that is matched to by $i \to v$ is already matched to by $j \to v$, and from (1) we know it does so cheaper, hence we get no gain by adding the shortcut $i \to v$.

$\square$

## A.4 Consistency

*Proof.* Here we will show that $L(S^n, \Theta)$ asymptotically behaves like *BIC*. $L(S, \Theta)$ directly corresponds to the log likelihood, which we rewrite as $\log p(S^n | \Theta, G)$ Our approach can be instantiated with arbitrary delay distribution, to show consistency we have to upper bound the number of parameters by $\mathcal{O}(\log n)$, this trivially holds for the parametric setting we focus on in this paper, because $|pa(i)| \in \mathcal{O}(\log n)$ [27]. The encoding of the graph $G$ is independent of $n$, i.e. fixed for a given network, hence in $\mathcal{O}(1)$. Finally this results at,

$$\log p(S^n | \Theta, G) + c \log n + \mathcal{O}(1)$$

we set $c = \frac{d}{2}$ where $d$ is the number of free parameters, arriving at the *BIC* score. $\square$

From Haughton [44] and Chickering [23] we know that *BIC* identifies a Markov equivalence class of the true DAG. For the identifiability of undirected edges we refer to Theorem 1 and 2. $\square$

## A.5 Connection to Hawkes Processes

The key difference between a linear Hawkes process and our model is the assumption of direct triggers, that is the mechanism of one event and one event only causing another. In a linear Hawkes process 'cause events' increase the intensity and therewith the probability of events occurring. However, one can generally not label for a specific event another as the 'cause', as each event is the result of a multitude of causes.

In this section, we are going to explore under which conditions we can identify a Hawkes process under our causal model.

Given an event sequence $S_j = \{t_k\}_{t_k=0}^{n_j}$ generated by a linear Hawkes process,

$$\lambda_j(t) = u_j + \sum_{i \in pa(j)} \sum_{t_k < t, t_k \in S_i} v_{i,j}(t - t_k) \ .$$

We construct a set of primary causes for each event $t_k \in S_j$ as

$$C_j = \left\{ (t, i, t_k) \mid t \in S_j, \ (i, t_k) = \arg\max_{(i, t_k), \ i \in pa(j), t_k < t, t_k \in S_j} v_{i,j}(t - t_k) \right\} \ ,$$

where we consider as primary cause of an event that past event with the highest influence at time point $t$ from a causal parent $i \in pa(j)$. Using these delays, we construct an alignment (mapping of delays) as

$$\Delta_{i \to j} = \{\beta(t_k) \mid t_k \in S_i\} \qquad \beta(t_k) = \begin{cases} t - t_k & \text{if } v_{i,j}(t - t_k) > u_j \\ \infty & \text{else} \end{cases},$$

where $t = \arg\max_{t \in S_j \wedge \exists(t,i,t_k) \in C_j} v_{i,j}(t - t_k)$ if $\nexists(t, i, t_k) \in C_j$ then $\beta(t_k) = \infty$. We consider $t_k \in S_i$ a cause of an event $t \in S_j$ if it is the primary cause of $t$ and $t_k$ has no stronger influence on any other event of $S_j$. Finally, the influence has to be stronger than that of the base intensity of $S_j$.

To be able to identify a causal edge between $S_i$ and $S_j$ the improvement gained by this alignment must outweigh the edge cost $L(i \to j)$. For this, there are two conditions: firstly, the number of primary cause events from $i$ to $j$, i.e. those instances where an event from $S_i$ has the maximum influence on an event from $S_j$, must be large enough. This is the case as long as $|\Delta_{i \to j}|$ increases with $n_j$, i.e. the total number of events of $S_j$. Then, in the limit $n_j \to \infty$ the number of primary cause events is large enough to offset the constant edge cost.

The achievable score gain is obtained by constructing a delay distribution from $|\Delta_{i \to j}|$ as

$$\phi_{i,j}(d) = \begin{cases} 1 - \alpha_{i,j} & \text{if } d = \infty \\ p_{\Delta_{i \to j}}(d) \cdot \alpha_{i,j} & \text{else} \end{cases} \qquad \text{where } \alpha_{i,j} = 1 - P(\Delta_{i \to j} = \infty).$$

If this density $\phi_{i,j}(d)$ fulfills the conditions of Theorem 2, we can identify $S_i$ as a parent of $S_j$ in the limit of $n_j \to \infty$.

In conclusion, CASCADE can identify a causal pair generated under a Hawkes process, if there exist sufficiently many events from $S_i \to S_j$, where $v_{i,j}(t)$ has the strongest influence on $\lambda_j(t)$ for some of the $t$. By aligning the delays of these events, we can identify the causal edge and recover the underlying causal structure.

## A.6 Empirical Evaluation

[From main paper] To empirically evaluate how effectively CASCADE recovers the true DAG on data generated by a Hawkes process, we generate synthetic data using the *tick* library[5]. We vary the intensity of the excitation function, i.e., the expected number of events generated per cause. We show the results in Figure 6. We observe that CAS-CADE performs best when our assumptions hold, when there is one effect per cause or fewer, but still demonstrates strong performance across all settings.

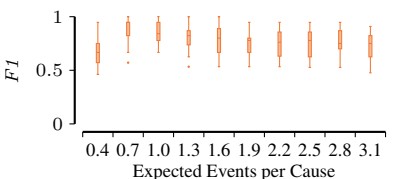

Figure 6: DAG recovery on data generated by a Hawkes process.

## A.7 Consistency of Algorithm

**Theorem 4.** *Given an event sequence $S$, where each individual subsequence $S_i$ was generated as per Eq. (3) by an underlying causal graph $G^*$. Assuming all $\Delta_{i \to j}$ are the true causal matchings. Under the Algorithmic Markov Condition, CASCADE recovers the true graph $G^*$ for $n \to \infty$.*

*Proof.* We begin by proving that in the first step, CASCADE identifies a true source node of $G^*$. We denote the parents of $i$ in the true causal graph $G^*$ as $pa(i)$, whilst we write for the parents in the graph maintained by CASCADE as $pa'(i, G)$.

Let $i \in [p]$ be a node of $G^*$ without parents, i.e. a source. By Theorem 3, for a path $i \to \cdots \to v \in G^*$ the gain in the causal direction $g(i \to v \mid \Theta)$ is greater than the gain in the reverse direction $g(v \to i \mid \Theta)$.

For all nodes $v \in [p], v \neq i$, $v$ is either an descendant of $i$ or unrelated.

1. If $v$ is a descendant of $i$, then the gain of $g(i \to v \mid \Theta)$ is greater than the gain of $g(v \to i \mid \Theta)$.

2. If $v$ is unrelated to $i$, then the gain in both sides is 0.

---
[5]`https://x-datainitiative.github.io/tick/`

Hence it follows, that for a node $i$ where $pa(i) = \emptyset$,

$$\max_{j \in C} g(j \to i \mid \Theta) - g(i \to j \mid \Theta) = 0 .$$

On the other hand, consider a node $i$ with parents $pa(i) \neq \emptyset$. Then, as $G^*$ is a DAG, there exists an ancestor $v$ of $i$, where $v$ is a source node, i.e. $pa(v) = \emptyset$. For that $v$, it holds that the gain from $v$ to $i$ is greater than the gain from $i$ to $v$. Hence, for a node $i$ with parents, it holds that

$$\max_{j \in C} g(j \to i \mid \Theta) - g(i \to j \mid \Theta) > 0 .$$

Therefore, by taking the argmin over all nodes, CASCADE identifies the true source node of $G^*$, i.e.

$$\arg \min_{i \in C} \max_{j \in C} g(j \to i \mid \Theta) - g(i \to j \mid \Theta) \implies pa(i) \cap C = \emptyset .$$

**Edge Addition**   Now, we show that CASCADE always identifies a true causal edge for $n_i \to \infty$. First, note that given a source node $i$, there do not exist any incoming causal edges in the graph $G^*$,

$$pa(i) \cap C = \emptyset \implies \nexists j \in C : j \to i \in G^* .$$

Hence, by fitting outgoing edges only, we test all possible edges for $i$ and never add a false oriented edge,

$$\forall j \in C : i \to j \in G^* \implies j \to i \notin G .$$

Finally, we recall that the true causal graph $G^*$ is the graph that minimizes the description length of the data as per the Algorithm Markov Condition. Hence, adding a true causal edge to the graph will result in a lower description length, i.e.

$$\forall j \in C, i \to j \in G^* : L(S_j | S_{pa'(j,G)}, \Theta') > L(S_j | S_{pa'(j,G) \cup i}, \Theta' \cup \theta_{i,j}) .$$

**Edge Removal**   Consider the node $i$, where $pa(i) \cap C = \emptyset$, and given a graph $G$ where all true causal edges have been added, i.e.

$$\forall j \in \bar{C} : \forall j \to v \in G^* : j \to v \in G .$$

Then, for $i$ it holds that

$$\forall j \in pa(i) : i \to j \in G .$$

It follows, that $pa'(i, G) \supseteq pa(i)$. By the Algorithm Markov Condition, the shortest description length of the data is achieved by the true causal graph $G^*$. Hence, a superset of the true parents of $i$ will result in a higher description length, and it holds that

$$L(S_i | S_{pa'(i)}, \Theta') > L(S_i | S_{pa(i)}, \Theta) .$$

Therefore, by testing that subset of the parents of $i$ results in a lower description length, CASCADE identifies the true parents of $i$.

**Overall Consistency**   For $n_i \to \infty$, we note that in each step for the node $i$ it holds that

1. $i$ has no parents in the candidate set $pa(i) \cap C = \emptyset$.

2. We add no false oriented edges to the graph, as $pa(i) \cap C = \emptyset \implies \nexists j \in C : j \to i \in G^*$.

3. We add all true edges $i \to j$ to the graph $G$, i.e. $\forall j \in C, i \to j \in G^* : L(S_j | S_{pa'(j,G)}, \Theta') > L(S_j | S_{pa'(j,G) \cup i}, \Theta' \cup \theta_{i,j})$.

4. For $i$, the current graph $G$ contains a superset of all true parents, i.e. $pa'(i, G) \supseteq pa(i)$, while the description length of the data is minimized by the true graph $L(S_i | S_{pa'(i)}, \Theta') > L(S_i | S_{pa(i)}, \Theta)$.

Hence, by repeating the edge addition and pruning in a topological order, in the limit of $n_i \to \infty$ under our causal model and by the Algorithm Markov Condition, CASCADE identifies the true causal graph $G^*$. $\qquad \square$

# B  Experiments

In this section we provide additional detail on the synthetic data generation and the experiment setup. Additional we provide further metrics on the synthetic experiments. For the real-world data we provide additional results.

## B.1  Synthetic Experiments

We generate synthetic data according to our causal model. We discretize the timestamps to 1 million unique timestamps. Throughout the experiments we vary the following parameters:

- Variables: Number of unique events types $p$.
- Edges: The total number edges in the generating causal graph $G*$.
- Delay Distribution: For all synthetic experiments we generate delays according to geometric distribution (as a discretize instantiation of the exponential).
- Delay Distribution Parameter: For each causal edge we sample the rate $\lambda$ uniformly from a specified range.
- Cause probability: For each causal edge we sample $\alpha$ uniformly from a specified range.
- # Source Events: Number of events sampled for source nodes (variables without any parents in the DAG).
- Additive noise parameter: percentage of additional added events to the caused events, also applies to source nodes, where # Source Events are considered as 'caused'.
- Instant effect: Except for the 'Instant Effect' experiments no instant effects are created.

For all experiments, unless otherwise stated a random DAG is generated. And for each parameterization 20 independent samples are generated.

**Sanity Check**   We set the number of types to 20 and generate 100 root events per source node (in this every node is a source node).

**Increase of Event Types**   In this experiment we increase the number of event types $p$ from 5 to 40, We set the number of edges to $(d^2 - d)/(2*5)$, that is 20% of all possible edges. To avoid overly many events in the colliders we set the number of root events to 20, for 40 variables this results in up to $\approx 30.000$ events. We do not include any additive noise and set $\alpha = 1$. For the delay distribution, we sample $\lambda$ from a range between of $[0.3, 1]$.

**Decrease of Noise**   In this experiment we increase the probability of $\alpha$, and decrease the fraction of additive noise. We set the number of variables to 20 and set the number of root events to 100. We sample $\lambda$ from a range of $[0.1, 0.4]$.

**Distribution Misspecification**   To further evaluate robustness of CASCADE we test recovery on generated data where the actual distribution does not match the assumed distribution. To this end, we change the assumed distribution of CASCADE and use the same setup as the previous experiment (Increase of Event Types) with 20 unique events. For the, true, exponential we observe an average F1 score of $0.82$, for the Poisson $0.81$, with a Normal distribution $0.76$, and uniform $0.75$. While recovery is best when assumed and generating distribution match CASCADE still performs well under misspecification.

**Multiple Parents**   For this experiment we specify a DAG, where $\lceil \frac{n-1}{2} \rceil$ are direct parents and $\lfloor \frac{n-1}{2} \rfloor$ are independent, the $n^{th}$ node is the collider. We plant 30 events per root cause and increase the number of variables from 50 to 200. We add 30 % of additive noise and set the cause probability randomly between 0.9 and 0.6. We repeat the same experiment where 10% of nodes are colliders. That is, for 50 event types, 5 are colliders and $\lceil \frac{p-5}{2} \rceil$ direct causes of all 5 colliders. The remaining $\lfloor \frac{p-5}{2} \rfloor$ are independent. We show the results in Table 1.

|  | F1 |
| --- | --- |
| Number of Colliders | |
| 5 | 0.97 $\pm$0.01 |
| 6 | 0.96 $\pm$0.01 |
| 7 | 0.95 $\pm$0.02 |
| 8 | 0.93 $\pm$0.01 |
| 9 | 0.92 $\pm$0.02 |
| 10 | 0.91 $\pm$0.01 |
| 15 | 0.88 $\pm$0.02 |
| 20 | 0.82 $\pm$0.01 |

Table 1: Average $F1$ score on *Multiple Parents* experiment with multiple colliders.

| | F1 | SHD | SID | SHD-Norm | SID-Norm |
| --- | --- | --- | --- | --- | --- |
| Method | | | | | |
| CASCADE | 0.74 | 19.45 | 104.10 | 0.05 | 0.27 |
| CAUSE | 0.19 | 264.50 | NaN | 0.69 | NaN |
| NPHC | 0.57 | 47.50 | NaN | 0.12 | NaN |
| THP | 0.23 | 64.20 | 180.70 | 0.16 | 0.47 |

Table 2: Average results on 90% instant data

**Instant Effects**    For the instant effects experiments we again use 20 variables with 100 root events, we shift the geometric delay distribution and set $\lambda = 0.9$, such that 90% of the events are generated at the same timestamp. We randomly sample the trigger probability between $0.7$ and $0.5$. For the exclusively instant effects we set $\lambda = 1$. We show the full results in Table 2 and

## B.2   Method Parameterization

**CASCADE**    We set the precision parameter for all experiments to 2. For all synthetic experiments we consider events as potential causes of at most 100 timestamps. For all experiments we consider a geometric distribution, which we shift back to cover instant effects.

**MDLH**    For the results of the *Increase event types experiment* we use the sparse version, where we set the maximum degree to the true maximal degree and set T =1000. In an effort to reduce runtime with higher number of types (i.e. nodes), we tested it with T=100, where it also did not terminate within 96 hours.

**Other**    For all other competing methods we used the default parameters.

## B.3   Compute Recourses

All experiments where executed on a internal cluster on compute nodes equipped with a AMD EPYC 7773X 64-Core Processor (2.2 GHz; Turboboost: 3.5 GHz), with 2 TB of RAM, while in practice a fraction of that was necessary. We provide the average runtimes below.

| | F1 | SHD | SID | SHD-Norm | SID-Norm |
| --- | --- | --- | --- | --- | --- |
| Method | | | | | |
| CASCADE | 0.55 | 32.80 | 216.05 | 0.08 | 0.56 |
| CAUSE | 0.19 | 249.60 | NaN | 0.65 | NaN |
| NPHC | 0.57 | 46.85 | NaN | 0.12 | NaN |

Table 3: Results of instant effects, we omit the results of THP as it only reports empty DAGs

| # Event Types | CASCADE | CAUSE | NPHC | THP | MDLH |
|---|---|---|---|---|---|
| 5 | 2.05 | 6.50 | 4.30 | 2.35 | 3.00 |
| 10 | 2.80 | 9.50 | 4.25 | 3.55 | 16489.00 |
| 15 | 7.10 | 16.75 | 4.40 | 13.80 | NaN |
| 20 | 35.20 | 42.95 | 4.15 | 40.90 | NaN |
| 30 | 362.65 | 225.70 | 4.75 | 305.50 | NaN |
| 40 | 1957.65 | 890.45 | 4.85 | 1984.60 | NaN |

Table 4: Mean runtime, in seconds, of Increase Event Types Experiment

| Noise | CASCADE | CAUSE | NPHC | THP |
|---|---|---|---|---|
| 0.10 | 147.85 | 231.50 | 4.55 | 97.20 |
| 0.50 | 66.85 | 176.85 | 4.40 | 71.45 |
| 0.60 | 42.75 | 142.30 | 4.50 | 61.55 |
| 0.70 | 25.20 | 113.30 | 4.40 | 51.35 |
| 0.80 | 14.05 | 76.75 | 4.25 | 41.65 |
| 0.90 | 7.55 | 48.60 | 4.45 | 32.95 |

Table 5: Mean runtime, in seconds, under increasing Noise.

## B.4   Network Alarms

In the provided dataset each event happens on a specific device. In addition to the event sequences a topology $\mathcal{T}$ over the devices is provided. An event can cause an event on each neighboring device, in addition to the device where the event occurred. To support this we can include a matching for connected devices. That is if $\{a, b\} \in \mathcal{T}$ we include $\Delta_{i \to j}^{(a,b)}$ and $\Delta_{i \to j}^{(b,a)}$, we include both directions since events on $a$ can cause events on $b$ and events on $b$ can cause events on $a$. For all devices we include the self loop $\Delta_{i \to j}^{(a,a)}$.

## B.5   Real World Experiment

In this section we provide Causal Graphs reported by CASCADE and for the *Global Banks* dataset additionally the result of THP.

| # Event Types | CASCADE | CAUSE | NPHC | THP |
|---|---|---|---|---|
| 50 | 45.80 | 323.65 | 6.30 | 587.45 |
| 60 | 79.60 | 536.75 | 6.10 | 1596.55 |
| 70 | 131.30 | 904.50 | 5.90 | 3175.30 |
| 80 | 196.80 | 1129.00 | 6.45 | 5393.70 |
| 90 | 283.90 | 1513.60 | 6.95 | 8295.25 |
| 100 | 392.55 | 1941.80 | 7.20 | 12923.50 |
| 150 | 1479.60 | 4694.75 | 10.05 | NaN |
| 200 | 3736.70 | 9736.05 | 16.95 | NaN |

Table 6: Mean runtime, in seconds, of Increase Event Types (Collider Experiment)

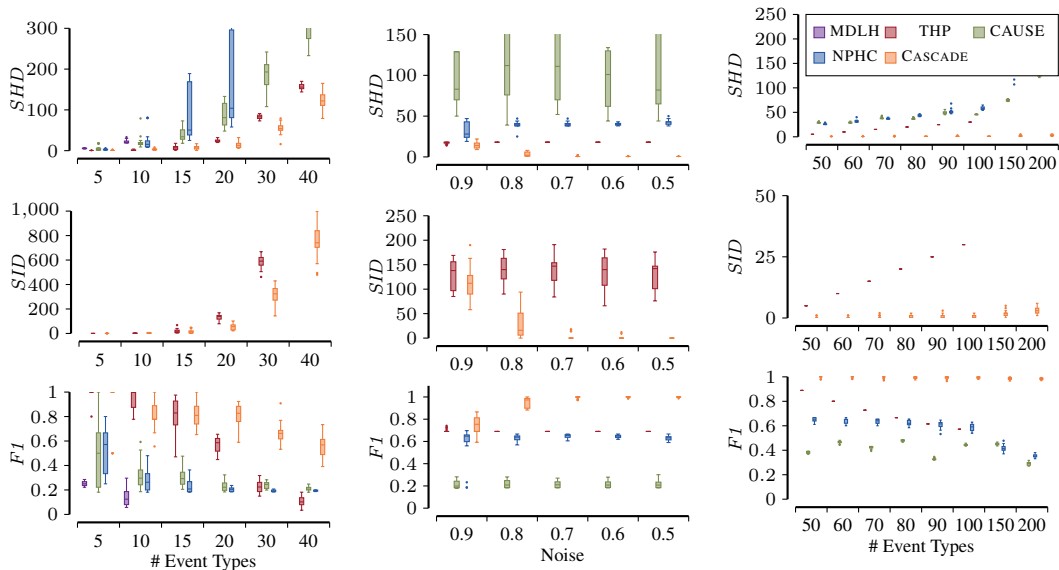

Figure 7: SHD, SID, and F1 score for the synthetic experiments

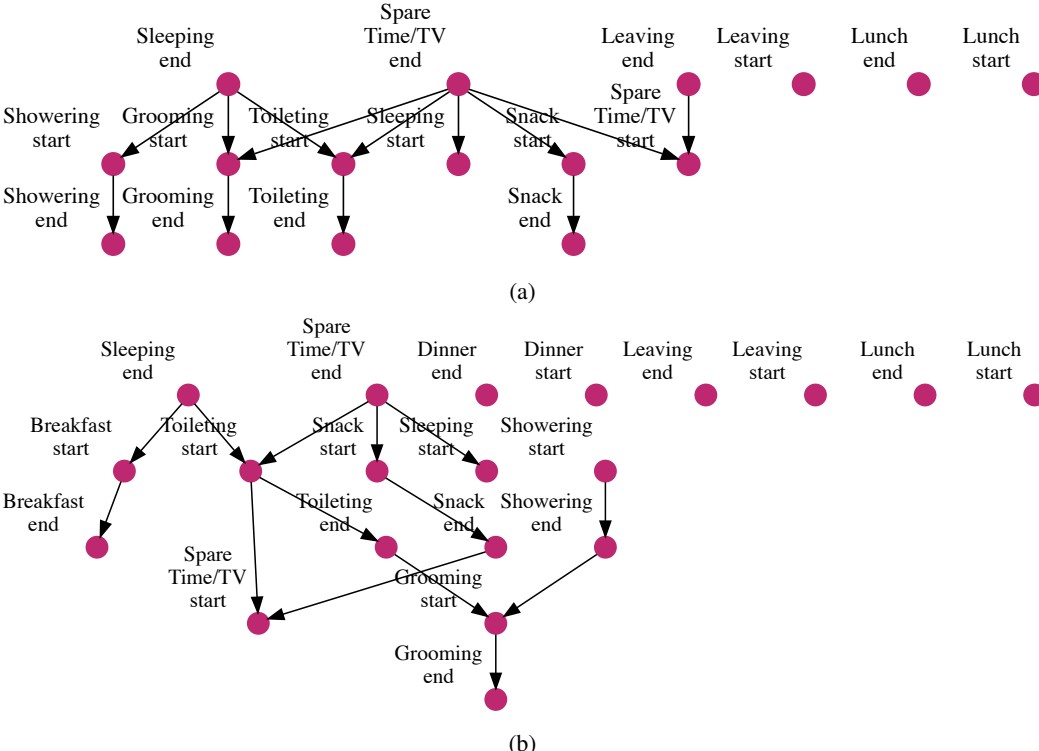

Figure 8: Recovered Causal Graphs on the two *Daily Activities* datasets.

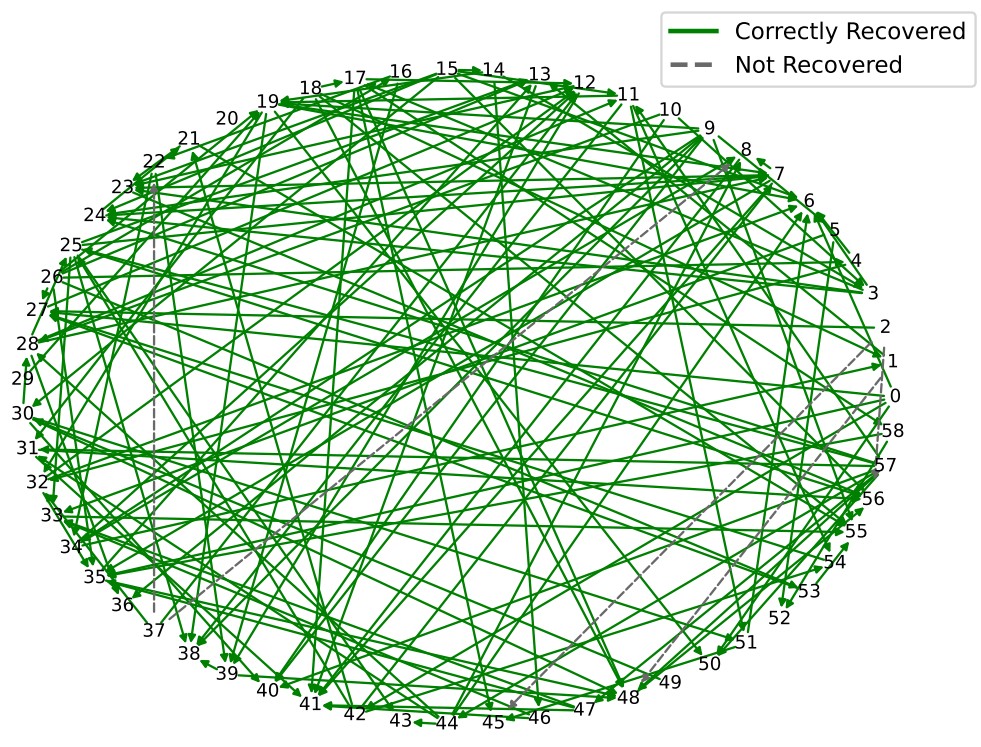

Figure 9: Recovered Causal Graph on Network Alarms dataset.

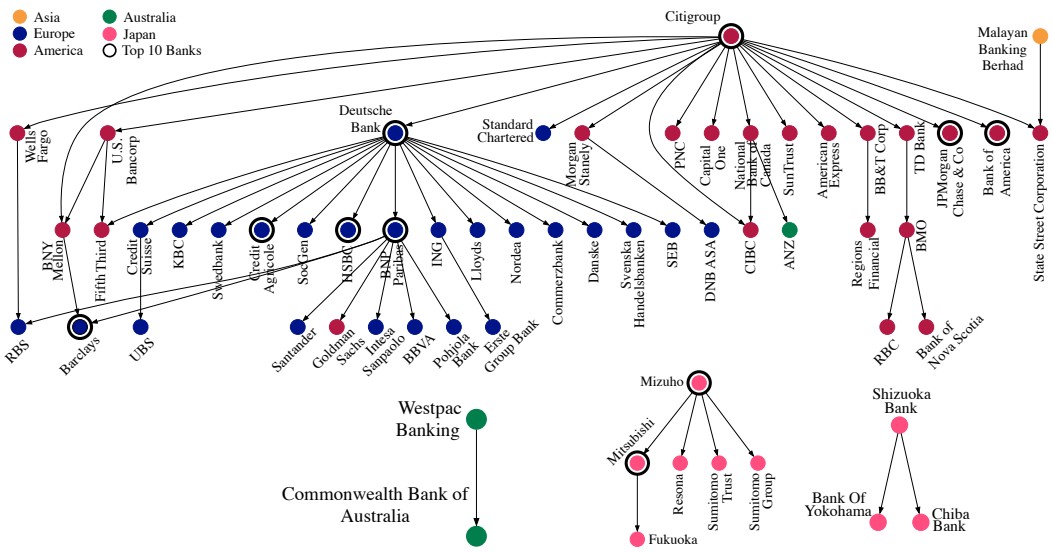

Figure 10: DAG reported by CASCADE on the *Global Banks* dataset [42]. We omit unconnected nodes for clarity.

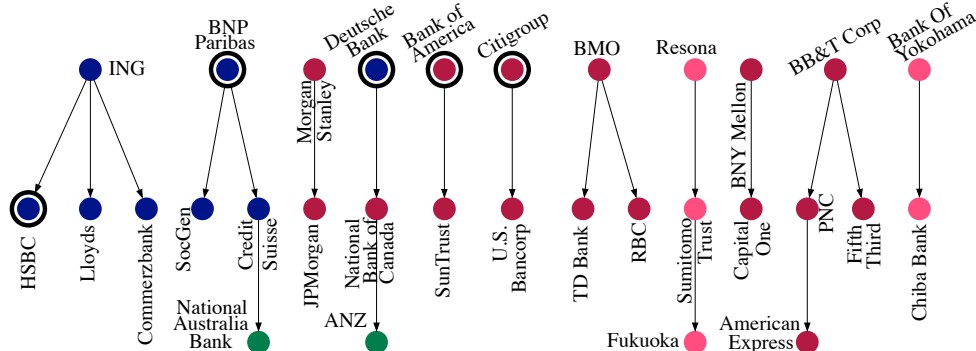

Figure 11: DAG reported by THP on the *Global Banks* dataset [42]. We omit unconnected nodes for clarity.

