# OpenReview forum: "Causal Discovery from Event Sequences by Local Cause-Effect Attribution"
_NeurIPS.cc/2024/Conference — NeurIPS 2024 poster_

### Official Review · Reviewer_e9Hj · 2024-07-10

**Soundness:** 3
**Presentation:** 3
**Contribution:** 3
**Rating:** 6
**Confidence:** 3

**Summary:**

This paper introduces a new causal model in which individual events of the cause variable trigger events of the effect variable with dynamic delays. The authors propose a cause-effect matching approach to learn a fully directed acyclic graph, named the CASCADE algorithm. The algorithm performs a topological search on observational data.

**Strengths:**

This paper presents a comprehensive theory and algorithm, and conducts extensive experiments, particularly with real data, to validate the effectiveness of the proposed method. The analysis and the algorithm are presented in a logical way.

**Weaknesses:**

The proposed method is the direct matching between a cause event and an effect event, which precludes modeling a single event causing multiple other events, as well as multiple events jointly causing a single effect event.  This limits the applicability of the algorithm.

**Questions:**

**1.**
Assuming ``an individual event ... causes an individual event'' in Line 64, and in Line 88: Equation 2.
Do these indicate that an effect event can only be caused by a single cause event? If so, the use of $pa(*)$ later in the manuscript is confusing. Can an effect event be caused by more than one event?

In the results of real experiments, there is an event that is caused by more than one event, which contradicts the assumption. How are the results obtained?


**2.**
Are timestamps in events erased during actual use?
As far as I can see, the algorithm proposed in the manuscript does not use timestamps.

**Limitations:**

The authors discuss in detail the limitations and applicability of their algorithm.

---

> ### Author Rebuttal · Authors · 2024-08-06
>
> Dear Reviewer, thank you for your time and valuable feedback.
>
> We want to follow-up on the applicability of our causal model and its implications. Mechanisms where multiple events are triggered, e.g. Hawkes processes, can still be modeled in parts by CASCADE. To this end, we supplement an experiment with a Hawkes process causal model,  where we vary the expected number of effect events following a cause. We provide the results in the Table below.
>
> |Events per cause | 0.4       | 0.7       | 1.0       | 1.3       | 1.6       | 1.9       | 2.3       | 2.6       | 2.8       | 3.1       |
> |-----------|-----------|-----------|-----------|-----------|-----------|-----------|-----------|-----------|-----------|-----------|
> | F1        | 0.69±0.13 | 0.89±0.11 | 0.86±0.10 | 0.80±0.12 | 0.79±0.12 | 0.76±0.09 | 0.75±0.11 | 0.75±0.12 | 0.75±0.11 | 0.74±0.11 |
>
> We observe that CASCADE suffers only a slight decrease in accuracy in the one-to-many (large number of expected events) and many-to-one cases.  In those settings, CASCADE models a subset of the true mechanism by matching the cause-effect events with highest likelihood. In general, for any causal model, CASCADE can identify causal edges as long as the fitted matching improves over the noise distribution. Expanding the causal model to many-to-one (A ∧ B => C) or suppression (A => !B) setting lies beyond the scope of this paper, but is nonetheless an interesting direction for future work, especially with regard to the identifiability of the resulting model.
>
> With regard to your questions:
>
> 1. **Multiple parents**: Each individual event, e.g. “C” occurring at $t_2$, can only be caused by a cause event of a particular kind, e.g. "A" at $t_1$. However, the entire sequence of “C” events can contain events caused by different parents. For example, assume that events of type "A" and "B" trigger an event of type "C". Given two events $(A,t_1)$ and $(B,t_3)$ , which cause $(C,t_2)$ and $(C,t_4)$ respectively, the resulting sequence $S_C = \lbrace t_2, t_4 \rbrace $ contains events from multiple parents, so that $pa(C) = \lbrace A, B \rbrace$. We hope this also clarifies the real-world results.
> 2. **Timestamps**: We use the timestamps when computing the edge cost $L(i \to j)$. We match events $t_i$ and $t_j$, so that the cost of the matched delays $d = t_i - t_j$ is minimized. We will clarify and expand the algorithm description in the updated manuscript.

---

> > ### Comment · Reviewer_e9Hj · 2024-08-12
> >
> > Thanks for your rebuttal. I will maintain my original score.

---

> > > ### Author Response · Authors · 2024-08-12
> > >
> > > Dear reviewer, thank you for your response! We are glad that our rebuttal answered your questions. If not please elaborate on your remaining concerns and aid us in addressing them adequately. Thank you very much!

---

### Official Review · Reviewer_WfU1 · 2024-07-10

**Soundness:** 3
**Presentation:** 3
**Contribution:** 3
**Rating:** 5
**Confidence:** 3

**Summary:**

The article employs the Algorithmic Markov Condition alongside Kolmogorov
complexity for causal discovery from event sequences. It focuses on a specific scenario in
which the sequence of events is divided into source and effect variables. The principal
contribution of this study is its innovative application of Pearl's causality model with
combination of AMC method, in contrast to the traditional Granger causality approach,
enabling the identification of both instantaneous and delayed effects.

**Strengths:**

1. Originality: The author employs Pearl's model of causality, diverging from
traditional Granger causality, to innovatively incorporate instantaneous effects
into the analysis of sequential events for causal relationship discovery.
2. Quality: The article is with good quality and honest about its strength and
limitation on their work.
3. Clarity: The article presents its algorithm with well-defined logic and
substantiated proofs.
4. Significance: The article offers an innovative approach to integrating
instantaneous effects into the causal discovery of sequential events, proposing a
potential method to enhance causal discovery techniques under such conditions.
However, it imposes strict limitations on the scenarios involving event sequences.

**Weaknesses:**

1. Significance: As mentioned in the limitation section by the author, strict
assumptions like direct matching between a cause event and an effect event leads
to challenges and possible violations in practical application, and it lacks
flexibility.
2. Section 3.3, which discusses the connection to Hawkes Processes, might be better
placed in an appendix or in a section dedicated to comparing different
methodologies. Its current placement in the theoretical part of the paper is
somewhat abrupt, especially since there is no direct focus on these processes in
your model.
3. The experimentation section lacks depth. It would be beneficial to evaluate and
report on the robustness of your model when its assumptions are challenged
during real-world applications.

**Questions:**

N/A

---

> ### Author Rebuttal · Authors · 2024-08-06
>
> Dear Reviewer, thank you for your time and valuable feedback. We would like to address your concerns first.
>
> 1. **Assumptions**: Mechanisms where multiple events are triggered, e.g. Hawkes processes, can still be modeled in parts by CASCADE. To this end, we supplement an experiment with a Hawkes process causal model,  where we vary the expected number of effect events following a cause. We provide the results in the Table below.
>
>    |Events per cause | 0.4       | 0.7       | 1.0       | 1.3       | 1.6       | 1.9       | 2.3       | 2.6       | 2.8       | 3.1       |
>    |-----------|-----------|-----------|-----------|-----------|-----------|-----------|-----------|-----------|-----------|-----------|
>    | F1        | 0.69±0.13 | 0.89±0.11 | 0.86±0.10 | 0.80±0.12 | 0.79±0.12 | 0.76±0.09 | 0.75±0.11 | 0.75±0.12 | 0.75±0.11 | 0.74±0.11 |
>
>      We observe that CASCADE suffers only a slight decrease in accuracy in the one-to-many (large number of expected events) and many-to-one cases. In those settings, CASCADE models a subset of the true mechanism by matching the cause-effect events with highest likelihood. In general, for any causal model, CASCADE can identify causal edges as long as the fitted matching improves over the noise distribution. Expanding the causal model to many-to-one (A ∧ B => C) or suppression (A => !B) setting lies beyond the scope of this paper, but is nonetheless an interesting direction for future work, especially with regard to the identifiability of the resulting model.
>
> 3. **Hawkes**: We will update the section on Hawkes processes to more closely show the connection to our causal model and reevaluate its proper placement in the paper.
> 4. **Experiments**: We evaluate our approach on real world data with unknown generating mechanisms. On labeled datasets (network alarms) CASCADE is far more accurate than the SOTA, whilst on the unlabeled global banks dataset we obtain a qualitatively better result than other methods. In both cases, our model is able to recover a sensible causal graph from the respective data distributions, even though it is unlikely that all of our assumptions hold.
>
>    In addition, we conducted in a controlled setting where we generate data outside of our causal model. Here, we examine CASCADE’s behavior under delay distribution misspecification. We report the results in the table below.
>
>    |    | Exponential | Poisson     | Normal      | Uniform     |
>    |-----|-------------|-------------|-------------|-------------|
>    | F1  | 0.82 ± 0.08 | 0.81 ± 0.09 | 0.76 ± 0.08 | 0.75 ± 0.10 |
>    | SHD | 13.8 ± 6.3  | 14.2 ± 6.7  | 18.5 ± 6.5  | 19.0 ± 7.8  |
>
>    While we observe a decrease in accuracy under misspecification, CASCADE still remains fairly accurate in detecting causal edges in the observed event sequences, as the misspecified model still provides an improved MDL-score over the noise distribution. In general, CASCADE’s is robust to different delay distributions, causal mechanisms and noise levels, and hence performant on the variety of data generating mechanisms found in the real world datasets.

---

> > ### Comment · Reviewer_WfU1 · 2024-08-12
> > **Thanks**
> >
> > Thanks for your rebuttal. I will maintain the original score.

---

### Official Review · Reviewer_T6V6 · 2024-07-11

**Soundness:** 2
**Presentation:** 3
**Contribution:** 3
**Rating:** 6
**Confidence:** 4

**Summary:**

In their work, the authors are concerned with recovering causal relations, where cause and corresponding effects occur in varying temporal distances. The authors leverage information theoretic formulations and properties of the algorithmic Markov condition to recover the causal graph via minimum description length principled. To this end, the authors present the 'CASCADE' algorithm, which recovers the topological ordering of the causal structure and proof identifiability results. The algorithm is evaluated on multiple synthetic data setups to examine the algorithm's performance under different varying noise, event type, and collider settings. Lastly, the algorithm is tested on a banking and daily activity data set to demonstrate robust performance on real-world data.

**Strengths:**

The paper is well-written and introduces the problem setup and formalisms intuitively. The authors consider the challenging problem of modeling causal event sequences. The information-theoretic treatise and causal modeling of the event-generating process via minimum description length encodings are well described and follow common notation from related work. While I am not an expert on the topic of time series event causality, relevant related work seems to be sufficiently discussed and compared to.

The overall intuition on all proofs is well described. To the best of my knowledge, proofs of theorems 1, 3 and 4 seem to be correct. (Please see minor comments on Thm. 2 below). The presented CASCADE algorithm seems to be sound and its robustness is evaluated via multiple real-world and synthetic experiments, varying the noise and number of event types.

**Weaknesses:**

While the authors present strong theoretical identifiability results, these guarantees are tied to a restrictive set of assumptions (faithfulness, sufficiency, low noise) and hold only for a specific type of event process (single excitation, no suppressing effects). While the authors state all assumptions explicitly, the paper could be improved by discussing the possible implications and reasonability of real-world applications.


Proof of Theorem 2 (Sec. A.2; second line of l. 496): As all other terms seem to be taken over from the line above, it is unclear to me where the canceled term on the left side of the inequality is coming from. (Since all terms are positive, I believe the transformation to be still correct.) Furthermore, it is not obvious to me how the equation following l.497 and the noise ratio of $n_{i,j}/n_j$ leads to the desired result. The paper could be improved by providing a more detailed explanation of this step.


The experiments seem to demonstrate consistently better results compared to related algorithms. However, from the experimental description in B.1, it seems that the experiment on the especially challenging identification of colliders --due to unclear parent assignment-- only considers a setting with a single collider. The authors might want to demonstrate algorithm performance for settings where multiple colliers exist, to better examine the algorithm's robustness regarding unclear EM assignments.


Minor:
* It would be helpful to mention the definition of H() in Sec. A.1 as the entropy, which is only mentioned afterward in A.2.
* Typos in the Proof of Thm. 2 (sec. A.2 l.490): "dealys", "ofset"; and the Conclusion (l.340) "discovers" -> "discover".
* In Sec. 4.1 l.201 text and formula disagree on the complexity: "[...] leading to an overall quadratic complexity $O(p^3)$".

**Questions:**

My questions mainly concern the weaknesses mentioned above. I would kindly like to ask the authors to comment on the following:

1) How realistic are the assumptions made in the paper (e.g., low noise in real-world settings)? How would one test for them to hold true? How robust would the algorithm be in the presence of other event types, such as suppressing events or multi-effect events?

2) Proof Thm. 2: Could the authors provide further details regarding the proof of theorem 2 - in detail, the derivation of the final step?

3) Regarding my comments above, could the authors give further insights on the algorithm's performance with an increased number of colliders?

4) Figures. 4, 6 and 8 seem to feature few colliders. This seems unreasonable to me, especially for the global banking data set, which I assume to be highly interconnected (possibly violating the DAG assumptions). Could the authors comment on this possible bias? Is it a result of the assumptions made, and how could it be reduced?

**Limitations:**

Limitations with regard to the applicability of the algorithm are discussed. Assumptions required for identifiability of the considered causal models are stated explicitly but might be hard to check in real-world settings. The work might be improved by discussing societal impacts from applying the algorithm under possible assumption violations in real-world settings.

---

> ### Author Rebuttal · Authors · 2024-08-06
>
> Dear Reviewer, thank you for your time and valuable feedback for the main paper as well as the Appendix. We would like to address your concerns and questions in detail.
>
> 1. **Assumptions**: First, we would like to elaborate on our assumptions and their implications.
>    - *Multiple effect events*: Mechanisms where multiple events are triggered, e.g. Hawkes processes, can still be modeled in parts by CASCADE. To this end, we supplement an experiment with a Hawkes process causal model,  where we vary the expected number of effect events following a cause. We provide the results in the Table below.
>
>       | Events per cause| 0.4       | 0.7       | 1.0       | 1.3       | 1.6       | 1.9       | 2.3       | 2.6       | 2.8       | 3.1       |
>       |-----------|-----------|-----------|-----------|-----------|-----------|-----------|-----------|-----------|-----------|-----------|
>       | F1        | 0.69±0.13 | 0.89±0.11 | 0.86±0.10 | 0.80±0.12 | 0.79±0.12 | 0.76±0.09 | 0.75±0.11 | 0.75±0.12 | 0.75±0.11 | 0.74±0.11 |
>
>      We observe that CASCADE suffers only a slight decrease in accuracy in the one-to-many (large number of expected events) and many-to-one cases. In those settings, CASCADE models a subset of the true mechanism by matching the cause-effect events with highest likelihood. In general, for any causal model, CASCADE can identify causal edges as long as the fitted matching improves over the noise distribution. Expanding the causal model to many-to-one (A ∧ B => C) or suppression (A => !B) setting lies beyond the scope of this paper, but is nonetheless an interesting direction for future work, especially with regard to the identifiability of the resulting model.
>    - *Low noise*: The low noise assumption is required only to identify exclusively instant effects. In Figure 3b) we provide the results of an experiment with both instant and delayed effects where the noise level is gradually increased. There, we observe that CASCADE works well even when the added fraction of noise events is at 90%.
>    - *Faithfulness, Sufficiency*: These assumptions are standard in causal discovery and allow us to show the fundamental identifiability of this mechanism for event sequences. For future work, it would be interesting to see if and how other MDL based approaches for confounding [1] and faithfulness [2] can be integrated into CASCADE.
>
>    [1] Kaltenpoth, David, and Jilles Vreeken. "Causal discovery with hidden confounders using the algorithmic Markov condition." Uncertainty in Artificial Intelligence. PMLR, 2023.
>
>    [2] Marx, Alexander, Arthur Gretton, and Joris M. Mooij. "A weaker faithfulness assumption based on triple interactions." Uncertainty in Artificial Intelligence. PMLR, 2021.
>
> 2. **Theorem 2**: We decompose the total number of events $n_j$ into those that were caused by variable $i$, i.e. $n_{i,j}$, and the remaining ones as $n_j - n_{i,j}$, which is the term that is canceled, i.e. $n_j = n_j - n_{i,j} + n_{i,j}$.
> We show that the entropy relation in (l.491) holds, since $\alpha_{j,j} n_j = n_{i,j}$ we can just replace $\alpha_{j,j}$ with $n_{i,j}/n_j$. We will clarify these steps in the updated manuscript.
>
> 3. **Colliders**: CASCADE is able to find collider structures on real world datasets. We provide the learned DAG of the Network Alarms dataset (Section 6.3) as Figure 1 in the rebuttal PDF. On this real world benchmark, CASCADE discovers many ground-truth collider structures. Additionally, we expanded the experiment from Section 6.2 to include graphs with multiple colliders (see Figure 2 in PDF). This setting too provides no problem for CASCADE, showing that our individual cause-effect matching correctly assigns effects to causes from different parents.
>
>    | # Colliders | 5        | 7        | 10       | 15      | 20       |
>    |-----|-----------|-----------|-----------|-----------|-----------|
>    | F1  | 0.97±0.01 | 0.95±0.01 | 0.91±0.01 | 0.87±0.02 | 0.82±0.01 |
>
> 4. **Global banks dataset**: Figure 8 shows the results of THP, which does not find any colliders at all, whereas CASCADE obtains a graph with causal edges from large to small banks with subgraphs representing geographical regions. We think, that the lack of collider structures stems from the fact that most crashes propagate on the same day. Hence, it is sufficient in most cases to have a causal singular causal parent, which is reflected in the graph we obtain in Figure 4. We agree that some of the assumptions are unlikely to hold, especially the DAG assumption and consider it an interesting future research direction to allow cyclic graphs.
>
>
> Regarding the detailed list of typos and misc suggestions, we thank you for reviewing our paper at this level of detail! We will adopt all proposed changes in the updated manuscript.

---

> > ### Comment · Reviewer_T6V6 · 2024-08-08
> >
> > Dear Authors,
> > thank you for answering my questions regarding the made assumptions and Thm. 2, as well as providing additional clarifying results on Hawkes process data and colliders.
> >
> > I still recommend the acceptance of this paper and will leave my score unchanged.

---

> > > ### Author Response · Authors · 2024-08-12
> > >
> > > Dear reviewer, thank you for your response! We are glad that our rebuttal answered your questions.

---

### Official Review · Reviewer_hRgg · 2024-07-12

**Soundness:** 3
**Presentation:** 2
**Contribution:** 3
**Rating:** 5
**Confidence:** 3

**Summary:**

The paper introduces a method for identifying causal relationships in event sequences. The authors presents a causal model that handles both instantaneous and delayed effects, contrasting it with existing methods like Granger causality. This algorithm is evaluated on both synthetic and real-world datasets.

**Strengths:**

1. The theoretical foundation based on the AMC and MDL principle is provided.

2. The proposed CASCADE algorithm is evaluated through extensive experiments.

3. The paper is well-organized, with clear explanations of the proposed model, theoretical underpinnings, and algorithmic steps. The use of illustrative examples and detailed proofs enhances understanding.

**Weaknesses:**

1. The paper acknowledges assumptions such as the direct matching between cause and effect events and the focus on excitatory effects. However, it could provide more discussion on the impact of these assumptions and potential ways to address them.

2. Scalability and computational complexity: The paper demonstrates the algorithm's performance on datasets with a moderate number of variables and events. An evaluation of its scalability to very large datasets, which are common in real-world applications, is less emphasized. The computational complexity of the algorithm, particularly for large datasets with many event types, is a concern. The quadratic complexity in the number of event types may limit its applicability to very large-scale problems.

3. Parameter sensitivity is not provided: How sensitive is the CASCADE algorithm to the choice of parameters for the delay distribution and cause probability?

**Questions:**

1. How sensitive is the CASCADE algorithm to the choice of parameters for the delay distribution and cause probability?

2. What are the practical limits of the CASCADE algorithm in terms of the number of event types and the size of the datasets?

3. How does the algorithm handle high levels of noise in the data, and are there specific noise thresholds beyond which performance degrades significantly?

**Limitations:**

The authors discussed the limitation in the Conclusion section. The identifiability of instantaneous effects relies on the strengths of the trigger and noise probabilities, which may be challenging to estimate accurately in practice.

---

> ### Author Rebuttal · Authors · 2024-08-06
>
> Dear Reviewer, thank you for your time and valuable feedback. We would like to address your concerns and questions in detail.
>
> 1. **Assumptions**: Mechanisms where multiple events are triggered, e.g. Hawkes processes, can still be modeled in parts by CASCADE. To this end, we supplement an experiment with a Hawkes process causal model,  where we vary the expected number of effect events following a cause. We provide the results in the Table below.
>
>    | Events per cause | 0.4       | 0.7       | 1.0       | 1.3       | 1.6       | 1.9       | 2.3       | 2.6       | 2.8       | 3.1       |
>    |-----------|-----------|-----------|-----------|-----------|-----------|-----------|-----------|-----------|-----------|-----------|
>    | F1        | 0.69±0.13 | 0.89±0.11 | 0.86±0.10 | 0.80±0.12 | 0.79±0.12 | 0.76±0.09 | 0.75±0.11 | 0.75±0.12 | 0.75±0.11 | 0.74±0.11 |
>
>    We observe that CASCADE suffers only a slight decrease in accuracy in the one-to-many (large number of expected events) and many-to-one cases. In those settings, CASCADE models a subset of the true mechanism by matching the cause-effect events with highest likelihood. In general, for any causal model, CASCADE can identify causal edges as long as the fitted matching improves over the noise distribution.
>    Expanding the causal model to many-to-one (A ∧ B => C) or suppression (A => !B) setting lies beyond the scope of this paper, but is nonetheless an interesting direction for future work, especially with regard to the identifiability of the resulting model.
>
> 2. **Scalability**: Scalability is an important aspect for application of causal discovery in real-world settings. In our experiments, we run with up to 200 variables (collider-experiment Figure 3 (c) ), which takes on average 62 minutes (single threaded), whilst the competitors closest in accuracy take 2.7 hours (CAUSE) and 3.6 hours respectively (THP). In the experiment with Hawkes processes, we process event sequences with 500k total events in 99 minutes. To obtain further speedup, the edge addition and pruning steps of CASCADE are fully parallelizable, which would allow our method to scale to extremely large problem sizes.
> 3. **Parameter sensitivity**: CASCADE does not require pre-specified parameters. The cause probability and delay parameters are fitted by minimizing the MDL score. CASCADE is sensitive to the decision which parametric model to adopt. Here, we choose the commonly used class of exponential distributions. To examine CASCADE’s behavior under misspecification, i.e. when the distribution is different, we conduct a further experiment. We report the results in the table below.
>
>    |     | Exponential | Poisson     | Normal      | Uniform     |
>    |-----|-------------|-------------|-------------|-------------|
>    | F1  | 0.82 ± 0.08 | 0.81 ± 0.09 | 0.76 ± 0.08 | 0.75 ± 0.10 |
>    | SHD | 13.8 ± 6.3  | 14.2 ± 6.7  | 18.5 ± 6.5  | 19.0 ± 7.8  |
>
>    While we observe a decrease in accuracy under misspecification that is worse for distributions of a different shape (normal, uniform), CASCADE still remains quite  accurate. In fact, CASCADE is able to detect causal edges in the observed event sequences, if the misspecified model still provides an improved MDL-score over the noise distribution.
>
>
> 4. **Noise**: Lastly, we would like to answer your question on the effect of noise on our method.
> The low noise assumption is only required to identify exclusively instant effects. For instant and delayed effects, Section 6.2 contains an experiment where the additional noise fraction of events is increased up to 90%. As summarized in Figure 3b), CASCADE copes well with noise and outperforms the SOTA on all noise levels.

---

> > ### Comment · Reviewer_hRgg · 2024-08-12
> >
> > Thanks the authors for the response. I will maintain my original score.

---

> > > ### Author Response · Authors · 2024-08-12
> > >
> > > Dear reviewer, thank you for your response! We are glad that our rebuttal answered your questions. If not please elaborate on your remaining concerns and aid us in addressing them adequately. Thank you very much!

---

### Author Rebuttal · Authors · 2024-08-06

We thank the reviewers for their detailed and thoughtful comments. All reviewers appreciate the proposed causal model with “theoretical foundation based on the AMC and MDL”, with “strong theoretical identifiability results”. In particular the identifiability of instant effects, which Granger causality based method can not identify, is recognized. Finally, the proposed CASCADE algorithm, with which “extensive experiments [...] to validate the effectiveness of the proposed method” were conducted, was well received across the board.

The reviews ask for clarification with regard to CASCADE’s performance in aspects regarding causal model, different data distributions and structure, and noise levels. We would like to respond to each question individually.
- **Causal model**: A commonly raised concern was the performance of CASCADE under a differing causal model, e.g. where one event causes multiple. To this end, we supplement an experiment with data generated by a Hawkes process, where we vary the expected number of effect events following a cause. We provide the results in the Table below.
   |Events per cause | 0.4       | 0.7       | 1.0       | 1.3       | 1.6       | 1.9       | 2.3       | 2.6       | 2.8       | 3.1       |
   |-----------|-----------|-----------|-----------|-----------|-----------|-----------|-----------|-----------|-----------|-----------|
   |F1        | 0.69±0.13 | 0.89±0.11 | 0.86±0.10 | 0.80±0.12 | 0.79±0.12 | 0.76±0.09 | 0.75±0.11 | 0.75±0.12 | 0.75±0.11 | 0.74±0.11 |

   We observe that CASCADE suffers only a slight decrease in accuracy in the one-to-many (large number of expected events) and many-to-one cases. In those settings, CASCADE models a subset of the true mechanism by matching the cause-effect events with highest likelihood. In general, for any causal model, CASCADE can identify causal edges as long as the fitted matching improves over the noise distribution.

- **Distribution misspecification**: In CASCADE, we fit exponential distributions to the matched cause-effect delays. We additionally test our method on generated data where the actual distribution is different and report the results in the table below.
   |     | Exponential | Poisson     | Normal      | Uniform     |
   |-----|-------------|-------------|-------------|-------------|
   | F1  | 0.82 ± 0.08 | 0.81 ± 0.09 | 0.76 ± 0.08 | 0.75 ± 0.10 |

   While we observe a decrease in accuracy under misspecification, CASCADE still remains accurate in detecting causal edges in the observed event sequences, as the misspecified model still provides an improved MDL-score over the noise distribution.

- **Multiple colliders**: CASCADE is able to find collider structures on real world datasets. We provide the learned DAG of the Network Alarms dataset (Section 6.3) as Figure 1 in the rebuttal PDF. On this real world benchmark, CASCADE discovers many ground-truth collider structures. Additionally, we expanded the experiment from Section 6.2 to include graphs with multiple colliders (Figure 2 in PDF). This setting too provides no problem for CASCADE, showing that our individual cause-effect matching correctly assigns effects to causes from different parents.

   | # Colliders | 5        | 7        | 10       | 15      | 20       |
   |-----|-----------|-----------|-----------|-----------|-----------|
   | F1  | 0.97±0.01 | 0.95±0.01 | 0.91±0.01 | 0.87±0.02 | 0.82±0.01 |

- **Strong noise**: The low noise assumption is only required to identify exclusively instant effects. For instant and delayed effects, Section 6.2 contains an experiment where the additional noise is increased up to 90%. As summarized in Figure 3b) of the paper, CASCADE copes well with noise and outperforms the SOTA on all noise levels.

In general, CASCADE’s is robust to different delay distributions, causal mechanisms and noise levels, and hence performant on the variety of data generating mechanisms found in the real world datasets. Expanding the causal model to the many-to-one (A ∧ B => C) or suppression (A => !B) setting lies beyond the scope of this paper, but is nonetheless an interesting direction for future work, especially with regard to the identifiability of the resulting model. We will update the manuscript with the provided experiments, as well as add a discussion about the implications of our assumptions to the paper.

---

### Decision · Program_Chairs · 2024-09-25

**Decision:**

Accept (poster)

**Comment:**

This paper advances the state-of-the-art in modeling event sequences as a way to gain insights into causation.  The reviewers are extremely consistent in their opinions.  The paper has a nice mixture of theoretical and empirical justification for its claims.  There was great discussion with the authors about questions around the following potential issues (1) assumptions (are they too strong?), (2) model misspecification, (3) multiple colliders, and (4) noise.  The authors gave additional empirical results, and these should be included in the final paper.  These better showed that the method had good robustness to different data distributions, structure, and noise levels.  They also showed the method could in fact handle situations where one event can influence multiple downstream events.  The reviewers seemed satisfied with the author replies although did not further revise their scores.  The authors should take into account reviewer comments and their own commitments in the replies to make clarifying additions in the final version.